# Monitoring Risks in Test-Time Adaptation

**Mona Schirmer**[1,*] **Metod Jazbec**[1,*] **Christian A. Naesseth**[1] **Eric Nalisnick**[2]
[1]UvA-Bosch Delta Lab, University of Amsterdam [2]Johns Hopkins University

## Abstract

Encountering shifted data at test time is a ubiquitous challenge when deploying predictive models. Test-time adaptation (TTA) methods address this issue by continuously adapting a deployed model using only unlabeled test data. While TTA can extend the model's lifespan, it is only a temporary solution. Eventually the model might degrade to the point that it must be taken offline and retrained. To detect such points of ultimate failure, we propose pairing TTA with risk monitoring frameworks that track predictive performance and raise alerts when predefined performance criteria are violated. Specifically, we extend existing monitoring tools based on sequential testing with confidence sequences to accommodate scenarios in which the model is updated at test time and no test labels are available to estimate the performance metrics of interest. Our extensions unlock the application of rigorous statistical risk monitoring to TTA, and we demonstrate the effectiveness of our proposed TTA monitoring framework across a representative set of datasets, distribution shift types, and TTA methods.

## 1 Introduction

Whenever test data is drawn from a different distribution than the one the model was trained on, performance might degrade, which can cause the model to 'expire'. Such drops in performance are especially concerning in safety-critical applications. For example, a medical device trained on patients from a specific demographic group may produce poor predictions when, upon deployment, it encounters patients from a different subpopulation.

Test-time adaptation (TTA) [44] has proven to be a powerful paradigm for prolonging the life of a model subjected to distribution shift. TTA methods adapt model parameters online, using only test batches of features and no labels. By leveraging unsupervised objectives such as test-time entropy [42] or pseudo-label losses [41], these methods effectively 'fine-tune' model parameters on unlabeled test data. Despite their stark potential to retain high performance under a variety of distribution shifts [49], TTA methods can suffer performance drops under severe shifts or after prolonged adaptation. The TTA literature has documented a range of persistent failure cases in which the model collapses entirely, resulting in near-zero accuracy [31]. Alarmingly, these failures often occur silently and prohibit TTA methods from safe deployment in practice.

Timely detection of performance degradations—whether due to harmful distribution shifts or adaptation collapse—is thus crucial for safe deployment. At the same time, however, falsely flagging that a model should be taken offline and retrained can incur significant, avoidable costs given the size of modern predictive models. Recently, sequential testing has emerged as a promising statistical framework for monitoring model performance over time [30]. When a predefined risk (or error) threshold is exceeded, the monitoring tool triggers an alarm. By employing time-uniform confidence sequences [13], such tools provide rigorous guarantees on the false alarm rate, thereby minimizing unnecessary retraining with high probability.

---

*Equal contribution. Corresponding authors: <`m.c.schirmer@uva.nl`, `m.jazbec@uva.nl`>

39th Conference on Neural Information Processing Systems (NeurIPS 2025).

However, existing sequential risk monitors either require access to ground truth labels in production [30] or do not account for model adaptation [1]. In this paper, we extend sequential testing to the challenging setting of TTA. This enables us to track the risk of a continuously evolving model *without ever observing test labels*. Our main contributions are as follows:

- In § 3, we present a general approach for risk monitoring of TTA models. Notably, our framework makes no assumptions about the distribution shift nor the TTA implementation.
- In § 3.2 and § A.1, we extend prior work on unsupervised risk monitoring [1] to enable effective tracking of risks most commonly used in TTA, such as classification error.
- In § 3.3 and § 3.4, we present a concrete instantiation of our monitoring tool based on model uncertainty, which importantly does not require fitting any additional model components.
- In § 5, we extensively study our monitoring tool and demonstrate that (i) it reliably detects risk violations and (ii) does not raise false alarms on a range of TTA methods, datasets and shift types.

## 2 Preliminaries

**Setting**  We consider a standard multi-class classification setting, where the input space is denoted by $\mathcal{X} \subseteq \mathbb{R}^D$ and the label space by $\mathcal{Y} = \{1, \ldots, C\}$ for some finite number of classes $C$. Data points $(\boldsymbol{x}, y)$ are assumed to be realizations of random variables $(\mathbf{x}, \mathbf{y})$ drawn from an unknown joint distribution $P$ over $\mathcal{X} \times \mathcal{Y}$. The samples in train $\mathcal{D}_{\text{train}}$ and calibration $\mathcal{D}_{\text{cal}}$ sets are drawn *i.i.d.* from the *source* distribution $(\boldsymbol{x}_0, y_0) \sim P_0$. Test samples in $\mathcal{D}_{\text{test}}^k$ are assumed to arrive *sequentially* from a time-varying and possibly shifting *test* distribution $(\boldsymbol{x}_k, y_k) \sim P_k$, $k \geq 1$. We do not make any assumptions about the nature of the distribution shift. For the test stream, we distinguish between two settings. In the 'unsupervised' setting, only test features $\boldsymbol{x}_k \sim P_k(\mathbf{x})$ are observed, yielding a sequence of unlabeled test datasets $\mathcal{D}_{\boldsymbol{x}}^k, k \geq 1$. In the 'supervised' setting, the true labels $y_k \sim P_k(\mathbf{y} \mid \mathbf{x} = \boldsymbol{x}_k)$ are revealed after predictions are made on the observed features at each time step $k$, resulting in a sequence of labeled test datasets $\mathcal{D}_{\boldsymbol{x}y}^k, k \geq 1$. Lastly, with $p : \mathcal{X} \to \Delta^{C-1}$ we denote a probabilistic classifier, where $\Delta^{C-1}$ is the probability simplex over $C$ classes.

**Losses and Risks**  It is crucial to monitor the deployed model on test data to detect potential performance degradations early. To formally capture the concept of error, a problem-specific *loss* function, denoted as $\ell : \mathcal{O} \times \mathcal{Y} \to \mathbb{R}$ is first defined.[2] The *risk* of a model $p$ on data drawn from distribution $P_k$ is then given as the expected loss $R_k(p) := \mathbb{E}_{P_k}[\ell(p(\mathbf{x}), \mathbf{y})]$. To ease notation, we denote the loss random variable on data from $P_k$ with $\mathbf{z}_k := \ell(p(\mathbf{x}), \mathbf{y})$ henceforth. $R_0(p)$ represents the *source risk* on data coming from the source distribution $P_0$. We also define a *running test risk* as

$$\bar{R}_t(p) := \frac{1}{t} \sum_{k=1}^{t} R_k(p) , \tag{1}$$

which measures the model's running performance on data drawn from the (shifting) test distribution $P_k$. If, for some time index $t^*$, the running test risk starts to exceed the source risk, i.e., $\bar{R}_{t^*}(p) > R_0(p)$, this may indicate that the data distribution has shifted in a way that is harmful to the model's performance, suggesting that the model should potentially be taken offline and retrained. In practice, the 'true' risk is typically estimated using the *empirical risk*, defined as $\hat{R}_k(p; \mathcal{D}_{\boldsymbol{x}y}^k) = \frac{1}{N_k} \sum_{n=1}^{N_k} z_{k,n}$, where the loss realizations $z_{k,n}$ are based on *i.i.d.* samples from $\mathcal{D}_{\boldsymbol{x}y}^k$.

Given our focus on classification, we consider two loss functions. The first is the 0-1 loss, $\ell_{0\text{-}1}(p(\boldsymbol{x}), y) := \mathbb{1}[\hat{y}(\boldsymbol{x}) \neq y]$ where $\hat{y}(\boldsymbol{x}) := \arg\max_c p(\boldsymbol{x})_c$, which means that the risk corresponds to classification error. We also consider a squared loss between labels and model confidences:

$$\ell_B(p(\boldsymbol{x}), y) := \frac{1}{2} \sum_{c=1}^{C} \left( p(\boldsymbol{x})_c - \mathbb{1}[y = c] \right)^2 . \tag{2}$$

When averaged across samples, this squared loss corresponds to the Brier score, a strictly proper scoring rule [9]. Hence it captures not only the classifier's error but also its calibration. Due to this connection, we refer to this second loss as the 'Brier loss' for short.

---

[2]The output space $\mathcal{O}$ may correspond either to the label space $\mathcal{Y}$ or to the space of probability distributions over $\mathcal{Y}$, depending on the loss type.

**Supervised Risk Monitoring via Sequential Testing**   To track how well a deployed model is performing, Podkopaev and Ramdas [30] propose a *risk monitoring* framework based on *sequential testing* [32]. The performance of a model $p$, in the presence of a *labeled* test stream, is tracked using the following sequential test:

$$H_0 : \bar{R}_t(p) \leq R_0(p) + \epsilon_{\text{tol}}, \ \forall t \geq 1 \qquad H_1 : \exists t^* \geq 1 : \bar{R}_{t^*}(p) > R_0(p) + \epsilon_{\text{tol}} \qquad (3)$$

where $\epsilon_{\text{tol}} > 0$ is a tolerance level that quantifies the acceptable drop in a model's test performance relative to its source performance.

To give the test anytime-valid properties (e.g. arbitrary stopping and restarting), Podkopaev and Ramdas [30] rely on *confidence sequences*, which extend traditional confidence intervals to the sequential setting and offer time-uniform coverage guarantees [5, 13]. A sequence of model losses on test data is used to construct an anytime-valid lower bound $L_t$ on the true running test risk $\bar{R}_t$:

$$\mathbb{P}\big(\bar{R}_t(p) \geq L_t(\mathbf{z}_1, \ldots, \mathbf{z}_t), \ \forall t \geq 1\big) \geq 1 - \alpha_{\text{test}}$$

for a miscoverage level $\alpha_{\text{test}} \in (0, 1)$. To get an upper bound $U$ on the source risk, a regular (static) confidence interval is computed using the loss on the source data:

$$\mathbb{P}\big(R_0(p) \leq U(\mathbf{z}_0)\big) \geq 1 - \alpha_{\text{source}}$$

for another miscoverage level $\alpha_{\text{source}} \in (0, 1)$. Combining the two bounds, the following *alarm function* is proposed

$$\Phi_t = \mathbb{1}\left[L_t(\mathbf{z}_1, \ldots, \mathbf{z}_t) > U(\mathbf{z}_0) + \epsilon_{\text{tol}}\right] \qquad (4)$$

and used to reject the null hypothesis (Eq. 3) at $t_{\min} := \inf\{t \geq 1 \mid \Phi_t = 1\}$. See Fig. 1 for an illustration. Note that in practice, the empirical bounds are computed using empirical risks:

$$\hat{U}_0 = \hat{R}_0(p; \mathcal{D}_{\text{cal}}) + w_0 \, , \quad \hat{L}_t = \frac{1}{t}\sum_{k=1}^{t} \hat{R}_k(p; \mathcal{D}_{\boldsymbol{xy}}^k) - w_t \, ,$$

where $w_0, w_t$ are finite-sample correction terms (see § B.1 for concrete formulas). Owing to the power of confidence sequences, the alarm function $\Phi_t$ enjoys a time-uniform guarantee on the control of the probability of the false alarm (PFA)

$$\mathbb{P}_{H_0}(\exists t \geq 1, \Phi_t = 1) \leq \alpha_{\text{test}} + \alpha_{\text{source}}$$

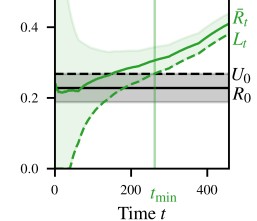

which ensures that performance degradations are not incorrectly detected, thereby avoiding unnecessary (and potentially costly) interventions on the model. Notably, this guarantee requires only that the loss function is bounded $\ell \in [a, b]$. This minimal assumption makes the method broadly applicable, as it imposes no constraints on the data distributions, the predictive model, or the nature of distribution shift (beyond independence). To maintain minimal assumptions, it is necessary to rely on a conjugate-mixture empirical Bernstein bound [13] when constructing a lower confidence sequence for the test risk $L_t$ (see § B.1 for more details).

Figure 1: Alarm $\Phi_t$ is raised at $t_{\min}$ as the lower bound $L_t$ on the running test risk $\bar{R}_t$ exceeds the upper bound $U_0$ on the source risk $R_0$.

**Test-time Adaptation (TTA).**   In test-time adaptation [42], the model parameters $\theta$ are updated online as the model observes batches of unlabeled test features. Specifically, given a sequence of unlabeled test batches $\mathcal{D}_{\boldsymbol{x}}^1, \ldots, \mathcal{D}_{\boldsymbol{x}}^t$, the TTA method produces a sequence of classifiers $p_{\theta_1}, \ldots, p_{\theta_t}$, where each $\theta_k$ is adapted using $\mathcal{D}_{\boldsymbol{x}}^k$. This stands in contrast to the static source model $p_{\theta_0}$, which is trained once on labeled training data $\mathcal{D}_{\text{train}}$ and remains fixed during deployment. For simplicity, we refer to the source model as $p_0$ and the adapted models as $p_1, \ldots, p_t$ henceforth.

## 3   Sequential Testing for TTA Monitoring

We now detail our approach to risk monitoring for test-time adaptation (TTA) methods using sequential testing. We begin by extending the risk monitoring framework of Podkopaev and Ramdas [30] to a deployment setting in which the model is continuously updated (§ 3.1). Next, inspired by Amoukou et al. [1], we derive a sequential test for the running test risk that does not require access to labels on the test data stream (§ 3.2)—a key innovation that enables rigorous statistical testing in TTA settings where test labels are unavailable. We then propose a concrete instantiation of our unsupervised test based on model uncertainty (§ 3.3) and online calibration of thresholds (§ 3.4). Finally, we describe techniques to enhance the detection power of the proposed tests (§ A.1). Our approach to monitoring risks in TTA is summarized in Algo. 1.

## 3.1 Risk Monitoring under Model Adaptation

Unlike in the static model setting considered by Podkopaev and Ramdas [30], we are interested in scenarios where a classifier is being continuously updated using a TTA method. Hence, we are interested in monitoring the risk not of a static source model $p_0$, but rather of a sequence of models $p_{1:t}$. To this end we define the hypotheses tested by our TTA risk tracker as:

$$H_0^a : \bar{R}_t(p_{1:t}) \leq R_0(p_0) + \epsilon_{\text{tol}}, \ \forall t \geq 1 \qquad H_1^a : \exists t^* \geq 1 : \bar{R}_{t^*}(p_{1:t}) > R_0(p_0) + \epsilon_{\text{tol}} \qquad (5)$$

where $\bar{R}_t(p_{1:t}) = \frac{1}{t} \sum_{k=1}^t R_k(p_k)$ and $R_k(p_k) := \mathbb{E}_{P_k}[\ell(p_k(\mathbf{x}), \mathbf{y})|\mathbf{x}_{1:k-1}]$. Note that conditioning on the (unlabeled) test stream $\mathbf{x}_{1:k-1}$ is included despite assuming an independent data stream, as it becomes necessary when the model $p_k$ is updated using test data, such as in TTA. To reduce notational clutter, this conditioning is omitted hereafter unless explicitly required. We use $\mathbf{z}_k^{(j)} := \ell(p_j(\mathbf{x}), \mathbf{y})$ to denote the loss random variable of the model $p_j$ on data from $P_k$.

To design the corresponding alarm function, we proceed similarly to the construction of $\Phi$ in Eq. 4, checking if the lower bound on the test risk exceeds the upper bound on the source risk. However, rather than relying on a sequence of losses from the static source model, we instead use a sequence of losses from the continuously adapted models in the (lower) confidence sequence for test risk. This leads to the adapted alarm function:

$$\Phi_t^a := \mathbb{1}\left[ L_t^a\big(\mathbf{z}_1^{(1)}, \ldots, \mathbf{z}_t^{(t)}\big) > U\big(\mathbf{z}_0^{(0)}\big) + \epsilon_{\text{tol}} \right], \qquad (6)$$

which also enjoys strong PFA control guarantees when using conjugate-mixture empirical Bernstein bounds [13]. Throughout the rest of this section, we abbreviate $\mathbf{z}_k^{(k)}$ as $\mathbf{z}_k$ and we shorten the sequence notation from $\mathbf{z}_1, \ldots, \mathbf{z}_t$ to $\mathbf{z}_{1:t}$ to ease the notational burden.

## 3.2 Unsupervised Risk Monitoring

While the adapted alarm function $\Phi^a$ (Eq. 6) monitors the performance of adapted models—rather than a fixed static model—it still depends on access to a labeled test stream to compute the adapted lower bound $L_t^a$. Consequently, it cannot be directly applied to track the performance of TTA methods where only an unlabeled test stream is available. To get around this, we propose to replace a sequence of supervised losses with a sequence of *loss proxies* that can be computed from unlabeled test streams. This allows us to derive an 'unsupervised' lower bound (Proposition 1) to the running test risk which we use to design an 'unsupervised' alarm function (Eq. 7).

As a first step, we introduce the notion of a loss proxy and specify its desirable properties. For a chosen proxy function $g$, a *loss proxy* of a model $p$ is defined as $\mathbf{u} := g(\mathbf{x}, p)$. Besides being 'unsupervised' (i.e., it should depend only on features $\mathbf{x}$), the proxy should be (at least partially) informative of the corresponding loss variable $\mathbf{z}$. Before presenting our concrete choice of a proxy function based on model uncertainty (see § 3.3), we formalize the notion of a proxy's informativeness with the following assumption.

**Assumption 1.** *Given a sequence of losses $\mathbf{z}_{0:t}$, let the corresponding sequence of loss proxies $\mathbf{u}_{0:t}$ and proxy thresholds $\lambda_0, \ldots, \lambda_t \in \mathbb{R}$, along with a loss threshold $\tau \in (0, M)$, be such that for all $t \geq 1$, the following inequality holds:*

$$\frac{1}{t} \sum_{k=1}^t \underbrace{\mathbb{P}_{P_k}\big(\mathbf{u}_k > \lambda_k, \mathbf{z}_k \leq \tau\big)}_{PFP_k} \leq \underbrace{\mathbb{P}_{P_0}\big(\mathbf{u}_0 > \lambda_0, \mathbf{z}_0 \leq \tau\big)}_{PFP_0} + \frac{1}{t} \sum_{k=1}^t \underbrace{\mathbb{P}_{P_k}\big(\mathbf{u}_k \leq \lambda_k, \mathbf{z}_k > \tau\big)}_{PFN_k}.$$

While Assumption 1 may initially appear rather complicated, it can be interpreted in terms of two intuitive desiderata for a valid (and effective) loss proxy. First, the proxy $\mathbf{u}$ should enable separation between low losses ($\mathbf{z} \leq \tau$) and high losses ($\mathbf{z} > \tau$) for a fixed loss threshold $\tau$. This ensures that the probabilities of both false positives ($PFP_k$) and false negatives ($PFN_k$) are small. Second, this separability should be robust across the time-varying test distributions $P_k$, ensuring that the false positive rate on the test stream ($PFP_k$) remains comparable to that on the source distribution ($PFP_0$). Below we formalize how a sequence of loss proxies can be used to derive an unsupervised lower bound on the true running test risk.

**Proposition 1.** *Assume a non-negative, bounded loss $\ell \in [0, M], M > 0$. Further, assume that for a sequence of losses $\mathbf{z}_{0:t}$, a sequence of loss proxies $\mathbf{u}_{0:t}$ together with thresholds $\lambda_0, \ldots, \lambda_t \in \mathbb{R}, \tau \in (0, M)$ satisfying Assumption 1 are available. Then the running test risk can be lower bounded as*

$$\bar{R}_t(p_{1:t}) \geq \underbrace{\tau \left( \frac{1}{t} \sum_{k=1}^{t} \mathbb{P}_{P_k}(\mathbf{u}_k > \lambda_k) - \mathbb{P}_{P_0}(\mathbf{u}_0 > \lambda_0, \mathbf{z}_0 \leq \tau) \right)}_{:=B_t}, \forall t \geq 1.$$

We defer the full proof to § B.2. A similar bound was proposed by Amoukou et al. [1], though with some key differences, which we discuss in detail in § A.1 and § 4. Importantly, the bound $\bar{B}_t$ from Proposition 1 depends only on the test loss proxies and the source loss, meaning its corresponding lower-bound confidence sequence $L_t^b$ can be evaluated using a combination of unlabeled test data ($\mathcal{D}_{\boldsymbol{x}}^k$) and labeled source data ($\mathcal{D}_{\text{cal}}$). This makes it suitable for our final proposed unsupervised alarm:

$$\Phi_t^b := \mathbb{1} \left[ L_t^b \big( \mathbf{u}_{0:t}, \lambda_{0:t}, \mathbf{z}_0, \tau \big) > U \big( \mathbf{z}_0 \big) + \epsilon_{\text{tol}} \right] . \tag{7}$$

In § B.3, we prove that such an alarm has a PFA control guarantee for the sequential test in Eq. 5.

### 3.3 Uncertainty as Loss Proxy

After introducing a general loss proxy $\mathbf{u}$ in the previous section, we now present a concrete instantiation based on model uncertainty. Specifically, we define the proxy function using the maximum class probability as $g(\mathbf{x}, p) := 1 - \max_c p(\mathbf{x})_c$. We choose uncertainty, firstly, due to it being easy to implement: it requires no modifications to the underlying model and avoids the need for additional components, unlike alternative proxies based on model disagreement [33] or separate error estimators [1, 4]. Secondly, for 0-1 loss this score approximates the conditional risk, up to calibration error:

$$R_{\text{0-1}}(p; \mathbf{x}) = \sum_{c=1}^{C} P(\mathbf{y} = c | \mathbf{x}) \cdot \mathbb{1}[\hat{y}(\mathbf{x}) \neq c] \approx \sum_{c=1}^{C} p(\mathbf{x})_c \cdot \mathbb{1}[\hat{y}(\mathbf{x}) \neq c] = 1 - \max_c p(\mathbf{x})_c .$$

Although the conditional risk approximation improves when $p$ is well-calibrated, we *do not* need to assume the model's uncertainty is well-calibrated under model adaptation [48] nor under distribution shift [28], as some previous work has required [16]. Returning to Assumption 1, uncertainty is a useful loss proxy when it separates high-loss and low-loss instances for a carefully chosen threshold $\lambda_k$—a task known as *failure prediction* [4, 50]. Failure prediction boils down to the ability to rank the test instances according to their loss values, which is a much weaker requirement in comparison to calibration [11]. Before demonstrating empirically in § 5 that using model uncertainty in Proposition 1 yields valid and tight lower bounds when monitoring TTA performance, we describe our threshold selection mechanism below.

### 3.4 Online Threshold Calibration

We now describe our procedure for selecting the loss and proxy thresholds used in the lower bound from Proposition 1. This step is critical for the effectiveness of our risk monitoring tool: poorly chosen thresholds can yield bounds that are either invalid (i.e., violate Assumption 1) or vacuous (i.e., excessively loose). Since our goal is to simultaneously minimize false positives and false negatives (cf. Assumption 1), we determine the loss threshold $\tau \in (0, M)$ and the source proxy threshold $\lambda_0 \in (0, 1)$ by maximizing the F1 score based on the source model's proxy:

$$\hat{\lambda}_0, \hat{\tau} := \arg\max_{\lambda, \tau} \text{F1}(\lambda, \tau; \mathcal{D}_{\text{cal}}, p_0) , \qquad \text{F1}(\lambda, \tau) = \frac{2\text{TP}}{2\text{TP} + \text{FN} + \text{FP}}$$

where $\text{TP} = \sum_{i=1}^{N_{\text{cal}}} \mathbb{1}[u_{0,i} > \lambda, z_{0,i} > \tau]$, $\text{FN} = \sum_{i=1}^{N_{\text{cal}}} \mathbb{1}[u_{0,i} \leq \lambda, z_{0,i} > \tau]$, $\text{FP} = \sum_{i=1}^{N_{\text{cal}}} \mathbb{1}[u_{0,i} > \lambda, z_{0,i} \leq \tau]$ and $u_{0,i} \sim \mathbf{u}_0$ and $z_{0,i} \sim \mathbf{z}_0$ are proxy and loss realizations of the source model $p_0$ on samples in $\mathcal{D}_{\text{cal}}$, respectively. Similarly, to select test proxy thresholds $\lambda_{1:t}$ we maximize F1 score while keeping the loss threshold $\hat{\tau}$ fixed: $\hat{\lambda}_k := \arg\max_\lambda \text{F1}(\lambda, \hat{\tau}; \mathcal{D}_{\text{cal}}, p_k)$, where F1 is computed from proxy $u_0^{(k)} \sim \mathbf{u}_0^{(k)}$ and loss $z_0^{(k)} \sim \mathbf{z}_0^{(k)}$ realizations of the *adapted* model $p_k$ on the (same) calibration dataset $\mathcal{D}_{\text{cal}}$ (since no test labels are available). We emphasize that continuously adapting the proxy threshold is essential for preserving an effective bound $\bar{B}_t$ under model adaptation. Using a static threshold throughout the test stream is insufficient, as many TTA methods can affect the scale of the observed uncertainties. For example, TENT [42] tends to reduce uncertainty over time due to its entropy minimization objective. Our full threshold selection procedure is summarized in Algo. 2.

# 4 Related Work

**TTA** [44, 20] aims to improve model performance under distribution shift by updating the model using unlabeled test data. Classic approaches include recomputing normalization statistics [36, 24], optimizing unsupervised objectives such as test entropy [42, 26, 27], energy [47], or pseudo-labels [17, 19], or adapting the last layer [15, 3, 35]. However, recent work has identified scenarios where TTA methods are ineffective [49, 35], and even harmful, degrading performance below that of the unadapted source model [3, 10, 46, 27, 6, 31, 43, 29]. While some studies propose heuristic indicators of TTA failure, such as high gradient norms [27], or estimate test accuracy directly [18, 31], there remains no principled framework for detecting risk violations of TTA methods with theoretical guarantees.

**Risk monitoring via sequential testing** has been proposed by Podkopaev and Ramdas [30], though in a setting where test stream labels are available and the model remains static. Recent important extensions include [51], which proposes supervised risk monitoring for the more challenging setting of instantaneous risk control, and [52], which tackles the task using weighted-conformal martingales. Most relevant to our work is that of Amoukou et al. [1], who extend [30] to the test scenario without labels. We further build upon their framework by: (i) incorporating model adaptation (§ 3.1); (ii) deriving an unsupervised bound on the expected loss, rather than only a bound on the probability of high loss (§ 3.2); (iii) using model uncertainty as a proxy for loss instead of a separate error estimator (§ 3.3); (iv) proposing a simpler calibration method (§ 3.4) and showing it's effectiveness for 0-1 loss (§ 5.1); and (v) providing theoretical insights into why effective monitoring of continuous losses necessitates a change in the tested hypothesis (§ A.1). Also related is work by Bar et al. [2], where a sequential test for TTA methods based on betting martingales [32] is proposed. However, their test is designed to detect changes in predictive entropy, which may or may not lead to a degradation in performance—unlike our method, which directly tests for performance drops. We provide further related works in Appendix E.

# 5 Experiments

We empirically validate the effectiveness of our monitoring tool for a range of TTA methods under different distribution shifts. In § 5.1, we study the monitoring tool in comparison to several baseline alarm functions. § 5.2 demonstrates the wide applicability of our monitoring tool across different TTA methods and datasets. In § 5.3, we show that the tool can be employed to detect risk increase arising from failed model adaptation. Lastly, in § 5.4, we show the generalizability of our statistical framework by going beyond uncertainty as loss proxy. Our code is available at: `https://github.com/monasch/tta-monitor`.

**Oracles and Baselines** Our goal is to approximate, $\hat{\bar{R}}_t := \frac{1}{t} \sum_{k=1}^{t} \hat{R}_k(p_k)$, the empirical estimate of the true, unobservable, running test risk $\bar{R}_t(p_{1:t})$ as closely as possible. Once $\hat{\bar{R}}_t$ exceeds a pre-defined risk threshold, we wish to raise an alarm as early as possible. We compare our unsupervised alarm $\Phi^b$ (Eq. 7 and Eq. 8), to several baseline monitors. While all monitors use the same upper bound on the source risk $U_0$, they differ in their choice of the test risk lower bound. We next present the alternatives to our proposed test risk lower bound $L_t^b$:

- $\hat{L}_t^a$: the estimated confidence lower bound on the running test risk under model adaptation (see Eq. 6). This direct extension of Podkopaev and Ramdas [30] preserves false-alarm guarantees but observes labels at each time point. While inapplicable in the unsupervised TTA setting, it serves as an oracle baseline. Since $L_t^a \geq L_t^b$ (under Assumption 1) our alarm $\Phi_t^b$ can never trigger before this alarm, $\Phi_t^a$, and consequently we inherit its detection delay. The closer $\hat{L}_t^b$ is to $\hat{L}_t^a$, the smaller is the price we pay for not observing test labels.
- $\hat{L}_t^c$: a naive estimate of the running test risk, formed by substituting the supervised losses $\mathbf{z}_{0:t}$ with unsupervised proxies $\mathbf{u}_{0:t}$ in the alarm from Eq. 4 [30]. While it avoids using test labels, it lacks false alarm guarantees due to omitting the lower bounding step in Prop. 1.
- $\hat{L}_t^d$: the estimated unsupervised lower bound on the running test risk of the static source model $p_0$ as presented in [1]. While providing false alarm guarantees without access to labels, it uses a different calibration procedure and is not applicable to a time-varying predictive model $p_k$.

**Risk Control Design**    We monitor test risk using 0-1 loss $\ell_{0\text{-}1}$ and Brier loss $\ell_B$ (Eq. 2). If not specified otherwise, we use a tolerance threshold of $\epsilon_{\text{tol}} = 0.05$ for 0-1 loss and $\epsilon_{\text{tol}} = 0.01$ for Brier loss. We set $\alpha = \alpha_{\text{source}} + \alpha_{\text{test}}$ to 0.2 using most budget for controlling the test risk, i.e. $\alpha_{\text{test}} = 0.175$ and $\alpha_{\text{source}} = 0.025$. For threshold selection (§ 3.4) we use $N_{\text{cal}} = 1000$ labeled samples from $P_0$.

**Datasets & Models**    We evaluate our monitoring approach on three datasets: synthetic corruptions from ImageNet-C [12], and real-world distribution shifts from Yearbook [8] and FMoW-Time [45]. For ImageNet-C, we use the pretrained ViT-Base model [7] from the Timm library [34], focusing on Gaussian noise (GN) corruptions. Yearbook involves binary gender classification from portrait images, while FMoW-Time consists of satellite imagery with land use labels. Both datasets span multiple years; models are trained on data up to a cutoff year and tested on future samples. For Yearbook and FMoW, we follow the protocol of Yao et al. [45], using their provided model weights: a small CNN for Yearbook and DenseNet121 [14] for FMoW.

## 5.1    Illustrative Example

In the first experiment, we study the behavior of our alarm in comparison to described baselines. We are notably interested how closely our unsupervised monitoring tool mimics the two oracle quantities having access to the ground truth test labels: empirical running test risk $\hat{\bar{R}}_t$ and $\hat{L}_t^a$ [30]. To simulate an increasing test risk, we construct a test stream from ImageNet-C by gradually increasing the severity level of Gaussian noise corruption from in-distribution (no shift) up to severity level 5. We track both the unadapted source model and TENT using 0-1 loss $\ell_{0\text{-}1}$ and Brier loss $\ell_B$. We also verify the validity of Assumption 1 throughout adaptation by tracking $\Delta_t^b = PFP_0 + \frac{1}{t}\sum_{k=1}^{t} PFN_k - PFP_k$. The assumption is met in practice when $\Delta_t^b \geq 0$ and violated when $\Delta_t^b < 0$. $\Delta_t^b$ also reflects the tightness of $L_t^b$, so, ideally, it is also not too much above 0. We proceed analogously for Assumption 4.1 in Amoukou et al. [1] and denote it with $\Delta_t^d$.

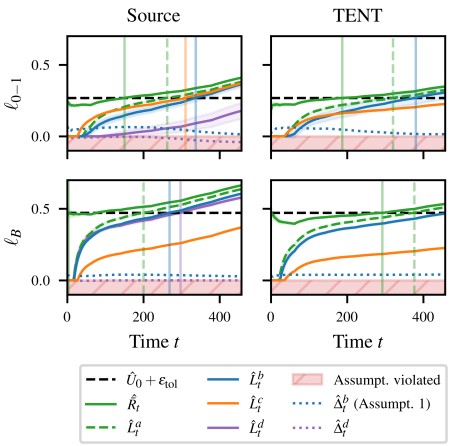

Figure 2: Test risk of increasing severity on ImageNet-C (GN): Our unsupervised lower bound $\hat{L}_t^b$ on the empirical test risk $\hat{\bar{R}}_t$ closely follows the supervised lower bound $\hat{L}_t^a$.

The results are shown in Fig. 2. As the severity of the distribution shift increases, the empirical running test risk $\hat{\bar{R}}_t$ (—) increases as well, for both 0-1 loss $\ell_{0\text{-}1}$ (*top row*) and Brier loss $\ell_B$ (*bottom row*). As expected, the unadapted source model (*left*) exhibits a higher risk, while adaptation with TENT (*right*) postpones the point where the empirical risk crosses the specified performance requirement. However, as the distribution shift becomes increasingly severe, $\hat{\bar{R}}_t$ eventually exceeds the upper bound on the source risk, $\hat{U}_t$, plus the tolerance margin $\epsilon_{\text{tol}}$ (- -), despite model adaptation. From this time point ( | ), we wish to trigger an alarm. As expected, the lower confidence sequence on the empirical test risk, $\hat{L}_t^a$ (- -), which leverages test labels, detects the risk violation first. Encouragingly, our proposed unsupervised lower bound $\hat{L}_t^b$ (—) closely follows the supervised bound $\hat{L}_t^a$. This indicates that our bound is tight and the price for not seeing labels is relatively small. The naive plugin bound, $\hat{L}_t^c$ (—), is not only void of theoretical guarantees but also exhibits low power empirically by not detecting the risk violation in all but one case. The unsupervised bound by Amoukou et al. [1], $\hat{L}_t^d$ (—), detects slightly later then $\hat{L}_t^b$ for Brier loss, but is extremely loose for 0-1 loss. Fig. 2 shows that our Assumption 1 (···) is met throughout the distribution shift in all cases, while the assumption of Amoukou et al. [1] (···) is violated for 0-1 loss, making their bound invalid for large $t$.

## 5.2    Generalization across Datasets, Shifts and TTA Methods

Next we evaluate the robustness of our monitoring tool by testing different TTA methods: TENT [42], CoTTA [41], SAR [27] and SHOT [19]. Please see § D.2 for details. We study four test streams: In-distribution of ImageNet (no shift, alarm should remain silent), ImageNet-C Gaussian

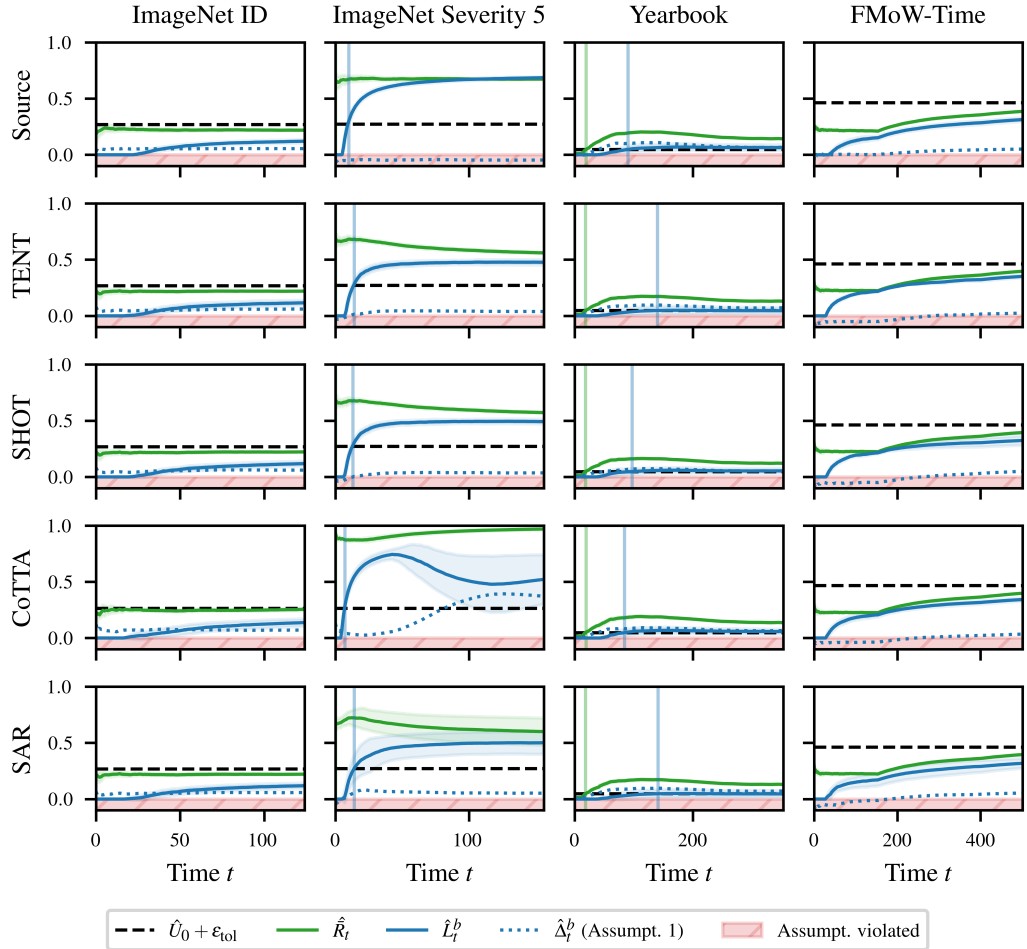

Figure 3: Estimated test risk for different datasets and TTA methods: Our lower bound $\hat{L}_t^b$ consistently exceeds the risk threshold $\hat{U}_0 + \epsilon_{\text{tol}}$ when a true risk violation occurs (ImageNet severity 5, Yearbook), while remaining below it on benign shifts (ImageNet ID, FMoW-Time), across all TTA methods.

noise (GN) severity level 5 (strong shift), Yearbook (moderate shift) and FMoW (gradual shift). Since classification error is the most commonly used metric in TTA, we track a risk increase for 0-1 loss.

**Risk violation is detected reliably** Fig. 3 shows that the empirical running test risk $\hat{\bar{R}}_t$ (—) is closely mimicked by our $\hat{L}_t^b$ (—) across TTA methods, datasets and shifts. Our alarm function correctly remains silent on the ID stream (*first column*) of ImageNet, where test risk remains below the threshold (- -). For FMoW, the risk increases steadily but also remains below the alarm threshold; this is accurately reflected in our monitoring, as $\hat{L}_t^b$ tightly tracks $\hat{\bar{R}}_t$ without triggering false alarms. For the immediate risk violation on ImageNet-C severity level 5 (*second column*), our test triggers an alarm after $< 25$ steps for all TTA methods. Similar results are observed for Yearbook. We additionally provide a detailed comparison with other baselines across all TTA methods in § A.4.

**Assumption 1 holds after warm-up** Importantly, we find that Assumption 1 is generally satisfied in practice, with $\hat{\Delta}_t^b$ (⋯) remaining above zero for most time steps, when using model uncertainty (§ 3.3) with online adaptation of proxy thresholds (§ 3.4). For some datasets, such as FMoW, we observe slight violations during the warm-up phase, i.e., for small $t$. Fortunately, the finite-sample penalty term $w_t$ in the confidence sequence is largest for small $t$, which may offset these minor violations and help prevent false alarms. The only instance where violations persist throughout the entire test stream is with the source model on ImageNet under a severity 5 shift. This is because our proposed threshold calibration procedure (§ 3.4) keeps the proxy threshold fixed if the model

is not updated on the test stream, making Assumption 1 more difficult to satisfy. Since our focus is on adapted models, we leave the development of alternative calibration methods effective for static models to future work.

## 5.3 Detecting TTA Collapse

Unlike static models, the risk of a TTA model can increase not only due to distribution shift but also because the model deteriorates during adaptation. An extreme, yet well documented case of model failure in TTA is model collapse, where finally only a small subset or a single class is predicted [25, 39, 27, 22]. Alarmingly, these harmful collapses often occur silently [27]. We next ask whether our statistical framework can detect risk increases caused by model failure. This is not a given, as the monitor relies on the model's own outputs (e.g., predictive uncertainty), which may become unreliable when the model itself fails. To induce model collapse, we follow [3] and apply TENT with a high learning rate of $\eta = 1e^{-1}$ on the ImageNet-C (GN) corruption at severity level 1. We set a high $\epsilon_{\text{tol}} = 0.2$ to disregard risk increase caused by distribution shift.

Fig. 4 (*left*) displays predicted classes (*first row*) and estimated test risk (*second row*) for an adaptation with a suitable learning rate. The predicted classes remain diverse, and both the estimated test risk and our lower bound

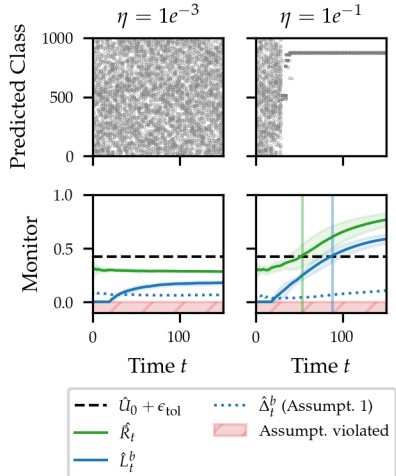

Figure 4: Collapsed vs. non-collapsed model on ImageNet-C (GN): When collapsed (*right*), the model always predicts the same class, which our monitor flags.

stay below the pre-defined risk threshold. This is in stark contrast to adaptation with a high learning (*right*): after few adaptation steps, the model assigns all input samples to the same class. This leads to a large increase in empirical test risk. Encouragingly, our bound $\hat{L}_t^b$ tracks this rise and detects a violation shortly after, demonstrating that our monitoring remains effective even when the underlying model collapses.

## 5.4 Alternative Loss Proxies

In § 5.2, we demonstrated that using model uncertainty as a loss proxy yields valid (according to Assumption 1) and, importantly, tight unsupervised lower bounds across a representative set of TTA methods and data shifts. Here, we supplement these results with a case where relying solely on model uncertainty proves insufficient for effective detection. Specifically, when monitoring "last-layer" TTA methods [15, 35]—which adapt only the classification head $W = [\boldsymbol{w}_1, \ldots, \boldsymbol{w}_C] \in \mathbb{R}^{H \times C}$—we observed that our unsupervised bound becomes overly loose, causing the alarm to fail under severe distribution shifts. We attribute this behavior to the normalization of

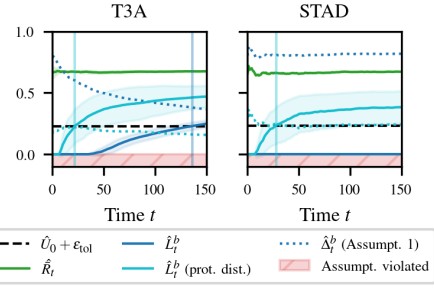

Figure 5: Comparison of loss proxies for last-layer TTA methods on ImageNet-C (GN) severity 5. Distance to class prototype is more effective than uncertainty for this TTA class.

each class prototype, i.e. $\boldsymbol{w}_c / \|\boldsymbol{w}_c\|_2^2$, at every adaptation step. This normalization leads to (much) reduced variability in uncertainty across samples at the start of adaptation, making the separation of high and low losses harder. To overcome this, we propose using the distance to the closest prototype [40, 23], $g(\mathbf{x}, p) = \min_c \|f(\mathbf{x}) - \boldsymbol{w}_c\|_2^2$, where $f$ denotes the feature extractor of model $p$, as an alternative loss proxy. Unlike uncertainty, this measure is less affected by the normalization of class prototypes. In Fig. 5, we show that this yields tighter lower bounds for T3A [15] and STAD [35]—both last-layer TTA methods—underscoring the importance of aligning the proxy choice with the specifics of the given TTA approach. Lastly, in § A.2 we show results when using energy score [21] or predictive entropy as a loss proxy—finding that both yield looser bounds compared to model uncertainty.

# 6  Conclusion

We proposed a risk monitoring tool for test-time adaptation (TTA) based on sequential testing. Crucially, our method is unsupervised—requiring no access to test labels—and is compatible with models undergoing continuous adaptation. We demonstrated its broad applicability across a diverse set of TTA methods, by showing that it effectively detects performance degradations resulting from either harmful distribution shifts or adaptation collapse.

**Limitations and Future Work**   While we have shown that our unsupervised alarm has detection delays not too much larger compared to its supervised counterpart [30] (see Fig. 2), the observed delays might still be too big for applications where detecting late is (significantly) more costly compared to raising false alarms. For such settings, it would be worth weakening the requirement on the probability of false alarm control under $H_0$ in order to gain more power under $H_1$. Perhaps this could be done by aiming for a weaker average run length control as is commonly done in the literature on change-point detection using confidence sequences [38, 37]. Moreover, although our proposed lower bound from Proposition 1 can be computed without access to test labels, verifying Assumption 1 for a given loss proxy still requires a labeled test stream. While we found empirically that this assumption holds in nearly all evaluated cases (Fig. 3), developing unsupervised diagnostics to flag potential violations of the assumption remains an important direction for future work.

## Acknowledgments and Disclosure of Funding

We thank Alexander Timans, Rajeev Verma, and Dan Zhang for helpful discussions and clarifications. This project was generously supported by the Bosch Center for Artificial Intelligence. Eric Nalisnick did not utilize resources from Johns Hopkins University for this project.

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

# Appendix

The appendix is organized as follows:

# A  Additional Results

## A.1  Improving Test Power

We have previously shown that our proposed unsupervised alarm (Eq. 7) provides strong false alarm control guarantees under $H_0$ (§ B.3)—that is, the alarm is guaranteed not to trigger when no performance degradation occurs in practice, preventing taking the model 'offline' prematurely. However, for a monitoring tool to be truly useful, it must also be 'reactive' under $H_1$—that is, it should raise an alarm when the model's performance degrades beyond an acceptable tolerance level ($\epsilon_{\text{tol}}$), and ideally, it should do so with minimal detection delay. Since our unsupervised alarm is based on lower-bounding the true running test risk twice —a lower-bound confidence sequence $L_t^b$ to a lower bound $B_t$ is used—it is not too surprising that the procedure can sometimes exhibit overly conservative behavior under $H_1$. We next discuss our strategies for addressing this issue by improving the power of our proposed unsupervised sequential test.

**0-1 loss**  We first note that for the 0-1 loss, the loss threshold $\tau$ can be omitted from the lower bound $B_t$ in Proposition 1. This is a direct consequence of the binary nature of the 0-1 loss (see Corollary 1 in § B.4 for the full derivation). Omitting this scaling for 0-1 loss yields a tighter lower bound $B_t$, which directly translates into a more reactive alarm function—while still maintaining false alarm guarantees. This is especially important when monitoring performance in TTA, where 0-1 loss is the one most widely used (as its risk corresponds to the classifier error).

**Continuous Losses**  For continuous losses such as Brier, which can take on any value in $[0, 1]$, the lower bound must be scaled by a threshold $\tau \in (0, 1)$, resulting in looser bounds.[3] To recover some of the lost test power, we propose also lower bounding the source risk $R_0$ using the same threshold $\tau$ as in the test lower bound $B_t$ (Proposition 1):

$$R_0 = \mathbb{E}[\mathbf{z}_0] \geq \tau \mathbb{P}(\mathbf{z}_0 > \tau) =: B_0$$

which follows directly from Markov's inequality. Denoting the corresponding upper-bound of the confidence interval for $B_0$ with $U^b$, we define the alarm function:

$$\Phi_t^\tau := \mathbb{1}\left[ \frac{1}{\tau} L_t^b\big(\mathbf{u}_{0:t}, \lambda_{0:t}, \mathbf{z}_0, \tau\big) > \frac{1}{\tau} U^b\big(\mathbf{z}_0, \tau\big) + \tilde{\epsilon}_{\text{tol}} \right] \tag{8}$$

---

[3]For a loss bounded in $[0, M]$ with $M > 1$, it is theoretically possible that threshold calibration yields $\hat{\tau} > 1$, in which case scaling by $\hat{\tau}$ could produce a tighter lower bound. However, since all losses considered in this work are bounded above by 1, we leave this case for future work.

where $\tilde{\epsilon}_{\text{tol}} := \frac{\epsilon_{\text{tol}}}{\tau}$ and show its PFA guarantee for the following sequential test

$$H_0^\tau : \frac{1}{t} \sum_{k=1}^{t} \mathbb{P}_{P_k}(\mathbf{z}_k > \tau) \leq \mathbb{P}_{P_0}(\mathbf{z}_0 > \tau) + \tilde{\epsilon}_{\text{tol}}, \ \forall t \geq 1 \tag{9}$$

$$H_1^\tau : \exists t^* \geq 1 : \frac{1}{t^*} \sum_{k=1}^{t^*} \mathbb{P}_{P_k}(\mathbf{z}_k > \tau) > \mathbb{P}_{P_0}(\mathbf{z}_0 > \tau) + \tilde{\epsilon}_{\text{tol}} \, .$$

The proof is provided in § B.5. Comparing the two sequential tests, we note that Eq. 9 tracks the probability of high loss, whereas Eq. 5 makes a statement about the *expected* loss (i.e., risk). While the test in Eq. 5 is arguably more interpretable—especially considering that the loss threshold $\tau$ is not specified by the user but determined empirically through a threshold calibration procedure (see Algo. 2)—the advantage of the high-loss probability test in Eq. 9 lies in its greater reactivity. Specifically, the lower bound $L_t^b$ in the alarm function (Eq. 8) is no longer scaled by $\tau$ (due to the multiplication by $\frac{1}{\tau}$), resulting in a tighter bound that can recover some of the statistical power lost in the continuous loss setting, albeit at the cost of reduced interpretability.

We also note that the scaled alarm function (Eq. 8) is closely related to the *quantile detector* proposed in Amoukou et al. [1]. Our work extends their approach in (at least) three main ways: first, by allowing for continuously evolving models, unlike [1] where a static model is assumed; second, by relaxing the assumption required for the loss proxy (see our Assumption 1 versus their Assumption 4.1); and third, by providing a theoretical justification for the increased reactivity of the high-probability test relative to the expected-loss test for continuous losses (via the cancellation of the loss threshold $\tau$). We elaborate further on these differences in § 4.

## A.2   Alternative Loss Proxies

Here, we compare our choice of using model uncertainty as a loss proxy (§ 3.3) with two alternatives: energy score and entropy. The energy score [21] is one of the most popular measures for detecting out-of-distribution (OOD) samples: $g(\mathbf{x}, p) = -\log \sum_{c=1}^{C} e^{m(\mathbf{x})_c}$, where $m(\mathbf{x}) \in \mathbb{R}^C$ is a vector of logits for model $p$. The entropy of the predictive distribution is another common uncertainty-based measure: $g(\mathbf{x}, p) = -\sum_{c=1}^{C} p(c|\mathbf{x}) \log p(c|\mathbf{x})$, where $p(c|\mathbf{x})$ is the predicted class probability. Entropy is widely used in TTA as an unsupervised objective, e.g., in TENT [42], where models adapt by minimizing predictive entropy on unlabeled test data.

We present the corresponding results in Fig. 6. While using energy as loss proxies also yields valid lower bounds (as indicated by $\hat{\Delta}_t^b > 0$), the resulting bounds (—) are consistently looser compared to those obtained when using model uncertainty (—) across all TTA methods and distribution shifts. We attribute the underperformance of the energy score to its focus on distinguishing in-distribution versus OOD samples rather than separating correctly and incorrectly predicted instances. In contrast, a useful loss proxy (see Assumption 1) should be able to differentiate between correctly predicted samples (i.e., low loss) and incorrectly predicted ones (i.e., high loss).

For entropy (—), detection delays are comparable to uncertainty as loss proxy. This confirms that our risk-monitoring mechanism remains valid and effective even when the same quantity is used both as the TTA objective and as the monitoring proxy. Indeed, our approach does not rely on the raw proxy values but instead on the proportion of test points exceeding a calibrated threshold (see Assumption 1). If adaptation minimizes entropy, our online calibration procedure (see § 3.4) dynamically adjusts this threshold, ensuring that the alarm remains reliable.

Although these findings further support our choice of model uncertainty as a suitable loss proxy, we believe that exploring alternative proxies that would lead to (even) tighter bounds remains an important direction for future work.

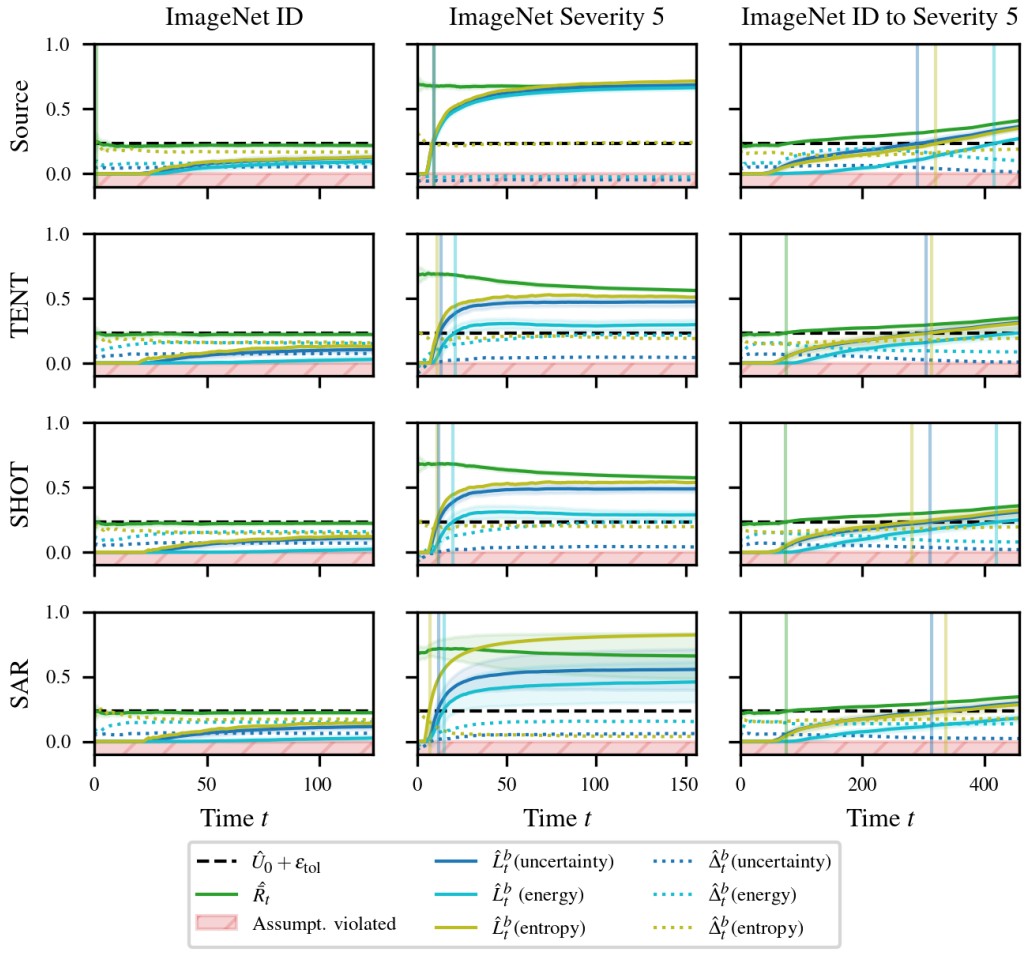

Figure 6: Estimated test risk for ImageNet test streams: We compare uncertainty and energy-score as loss proxies. Uncertainty yields consistently tighter lower bounds on the test risk than the energy score.

### A.3 Ablation on Calibration Set Size and Calibration Frequency

Our method requires a small labeled calibration set from the source distribution, which introduces additional computational overhead due to repeated evaluation of the adapted model on this set during adaptation. A small set of (labeled) source samples is commonly used for initializing TTA methods [53, 54, 41, 55, 56, 2, 57–59], and is indispensable for risk control [30, 1]. We next investigate whether the calibration set size can be reduced. In addition to addressing the reliance on labeled data, a smaller calibration set also reduces the runtime overhead of the online calibration procedure.

Fig. 7 shows the estimated upper bound on the source risk $\hat{U}_0$ and our estimated lower bound on the test risk $\hat{L}_t^b$ for the default calibration set size of $N_{\text{cal}} = 1000$ (—) and a reduced size of $N_{\text{cal}} = 100$ (—). We find that reducing the calibration set to $N_{\text{cal}} = 100$ has minimal impact on both $\hat{U}_0$ and $\hat{L}_t^b$. Only in the setting with a gradually increasing distribution shift (*right column*) do we observe a slight delay in risk detection compared to the default setup.

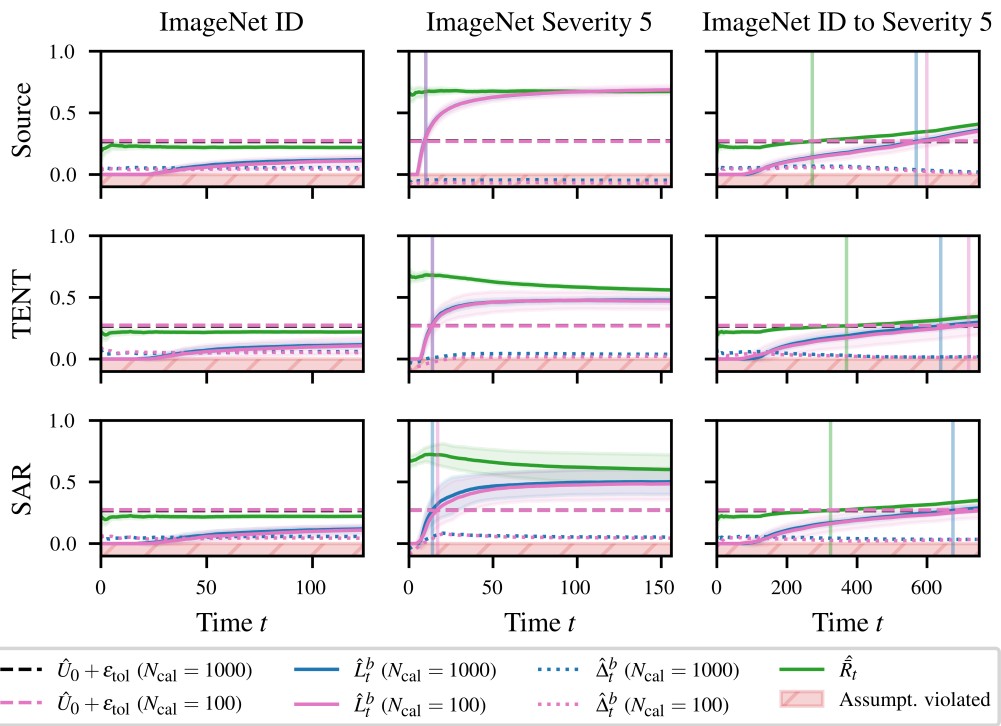

Figure 7: Estimated test risk for ImageNet test streams. We compare the default calibration set size of $N_{cal} = 1000$ to a smaller set with $N_{cal} = 100$. Reducing the calibration size leads to no or only small detection delays relative to the larger calibration set.

Another way to reduce the computational overhead of the monitoring tool is to decrease the frequency of the online calibration procedure described in § 3.4. Instead of evaluating the adapted model on the calibration set and selecting a new threshold $\lambda_k$ after every adaptation step $k$, one can perform this threshold selection only periodically, reusing the most recently calibrated threshold for the intermediate monitoring intervals. This reduces the number of calibration evaluations while maintaining continuous monitoring of the model's performance.

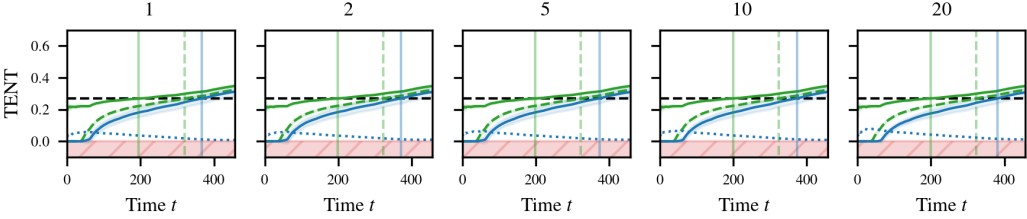

Figure 8: Estimated test risk of TENT on ImageNet-C (ID to Severity 5) when performing the online calibration step every $1, 2, 5, 10, 20$ adaptation steps. Less frequent calibration results in only a small increase in detection delay, indicating a good trade-off between detection delay and computational overhead.

To assess the impact of such reduced calibration frequency, we monitor TENT performance on the ImageNet-C test stream with different intervals—performing calibration every $1, 2, 5, 10, 20$ adaptation steps. Fig. 8 shows that less frequent calibration leads to only a small increase in detection delay. This suggests that our monitoring framework remains effective even when calibration is performed at coarser temporal resolutions. Future work may further explore adaptive calibration strategies that trigger re-calibration only when a significant increase in estimated risk is detected, rather than at fixed time intervals.

## A.4  Extended Comparison to Baselines

We next provide an extended baseline comparison by evaluating the baselines from § 5.1 on the TTA methods and datasets studied in § 5.2.

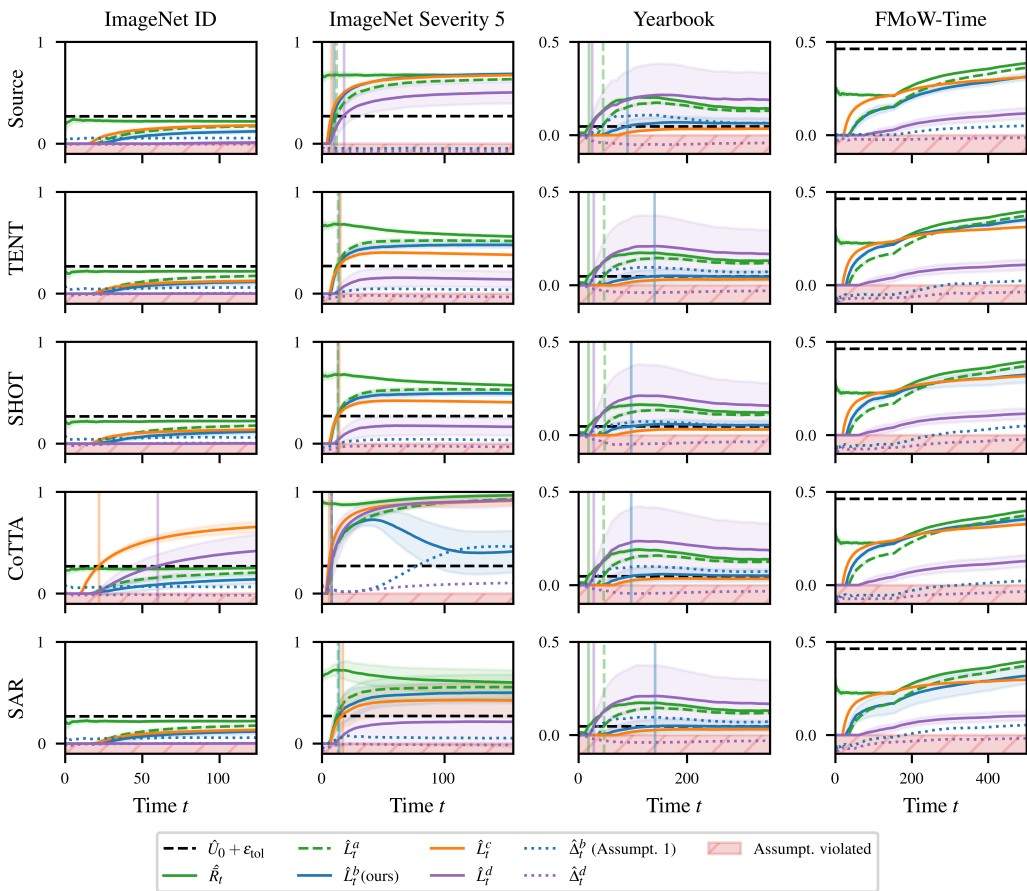

Figure 9: Estimated test risk for different baselines, datasets and TTA methods.

Fig. 9 displays the estimated test risk across datasets and TTA methods. As expected, the oracle, supervised lower bound on the test risk, $\hat{L}_t^a$ (- -), reliably flags risk violations without causing false alarms across all datasets and TTA methods. In contrast, the naive plug-in bound $\hat{L}_t^c$ (—) triggers a false alarm on the in-distribution ImageNet test stream for CoTTA, despite the test risk remaining below the alarm threshold. This is unsurprising, as $\hat{L}_t^c$ lacks formal guarantees on the false alarm rate. While it yields reasonable risk estimates for the other TTA methods on ImageNet ID, as well as on ImageNet-C severity 5 and FMoW-Time, it fails to detect risk violations on Yearbook across all TTA methods. Even though not originally proposed for TTA, we extend the unsupervised test risk lower bound by Amoukou et al. [1], $\hat{L}_t^d$ to the TTA setting to enable comparison on this plot. We note that it behaves poorly with TTA methods. $\hat{L}_t^d$ (—), also triggers a false alarm on ImageNet ID for CoTTA. For other TTA methods, it is largely unresponsive resulting in a consistently loose lower bound on the estimated true test risk $\hat{\bar{R}}_t$ (—). This looseness leads to missed alarms on the severe shift of ImageNet-C severity 5 for 3 out of 5 TTA methods. Furthermore, we observe that the required assumption of their method (⋯) is violated in nearly every practical setting.

In contrast, as shown in § 5.2, our unsupervised test risk lower bound $\hat{L}_t^b$ (—) detects risk violations promptly (ImageNet-C severity 5, Yearbook), while remaining inactive when the risk threshold is not breached (ImageNet ID, FMoW-Time).

## A.5 Results under Label Shift

As our theoretical framework does not make any assumptions about the nature of the distribution shift (see § 2), it naturally extends to scenarios with shifting label distributions. To explicitly validate this, we conducted additional experiments on the Yearbook dataset, where we induced controlled label shift by reordering test samples according to their class labels, following the setup of [10, 35].

Fig. 10 compares our monitoring results for Yearbook with (*right*) and without (*left*) label shift. Test-time adaptation methods perform worse under label shift, as reflected by the higher empirical risk $\hat{\bar{R}}_t$ (—). Our method reliably detects this deterioration, triggering an earlier alarm across all TTA methods. While Assumption 1 becomes slightly looser in this setting, the monitor still raises an alert, demonstrating that our framework effectively captures failures of TTA methods caused by label shift.

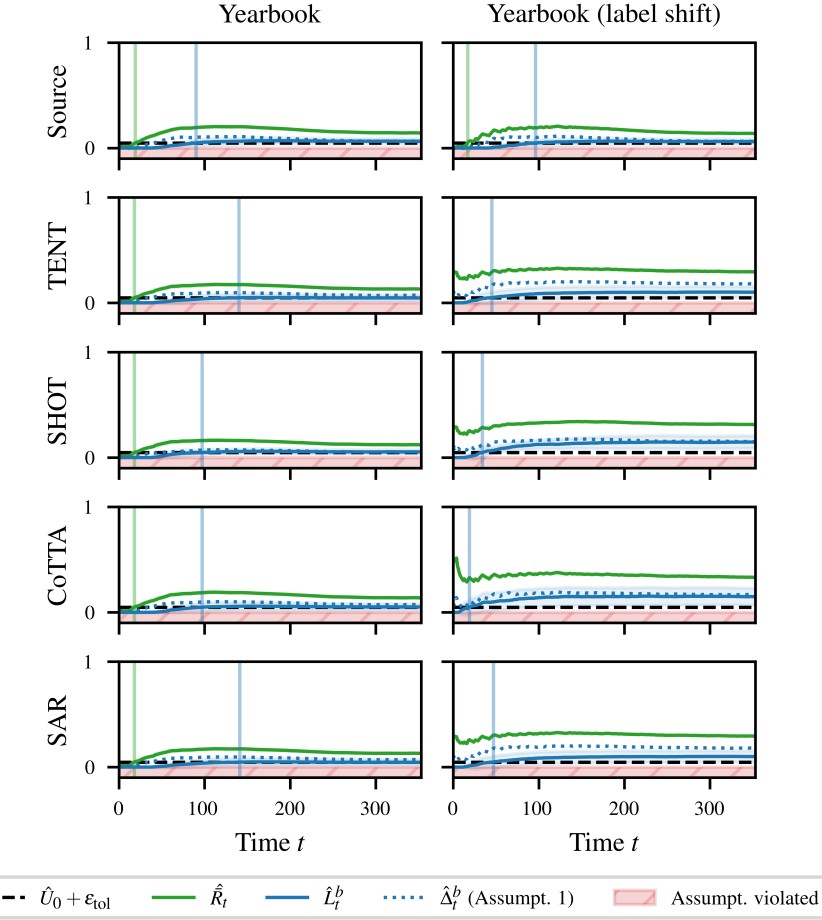

Figure 10: Estimated test risk on Yearbook. We compare a test stream affected only by covariate shift (*left*) with one that additionally exhibits label shift (*right*). Under label shift, TTA methods show higher empirical risk, while our risk monitor detects the degradation earlier and raises an alarm accordingly.

# B Theoretical Results

## B.1 Choice of Confidence Sequences

For an introduction to confidence sequences, we refer the interested reader to Howard et al. [13] and Appendix E of Podkopaev and Ramdas [30]. Throughout this paper, we use a Hoeffding confidence interval to estimate the upper bound on the source risk. Accordingly, the finite-sample penalty term is given by $w_0 = \sqrt{\log\left(1/\alpha_{\text{source}}\right)/N_{\text{cal}}}$. For the lower bound on the test risk, we use the conjugate-mixture empirical Bernstein (CM-EB) confidence sequence proposed in Theorem 4 of Howard et al. [13], chosen for its minimal assumptions. To obtain the finite-sample terms $w_t$ for $t \geq 1$—which depend on $\alpha_{\text{test}}$ and the empirical variance of the observed sequence—we use the gamma-exponential mixture bound from Proposition 9 in Howard et al. [13].

For the confidence sequence $L_t^b$ used in our proposed unsupervised alarms $\Phi_t^b$ (Eq. 7) and $\Phi_t^\tau$ (Eq. 8), we apply a CM-EB lower confidence sequence for $\frac{1}{t}\sum_{k=1}^t \mathbb{P}_{P_k}(\mathbf{u}_k > \lambda_k)$ with $\alpha_{\text{test}_1}$, and an upper Hoeffding confidence interval for $\mathbb{P}_{P_0}(\mathbf{u}_0 > \lambda_0, \mathbf{z}_0 \leq \tau)$ with $\alpha_{\text{test}_2}$, such that $\alpha_{\text{test}_1} + \alpha_{\text{test}_2} = \alpha_{\text{test}}$.

## B.2 Unsupervised Lower Bound Derivation (Propositon 1)

**Proposition 1.** *Assume a non-negative, bounded loss $\ell \in [0, M], M > 0$. Further, assume that for a sequence of losses $\mathbf{z}_{0:t}$, a sequence of loss proxies $\mathbf{u}_{0:t}$ together with thresholds $\lambda_0, \ldots, \lambda_t \in \mathbb{R}, \tau \in (0, M)$ satisfying Assumption 1 are available. Then the running test risk can be lower bounded as*

$$\bar{R}_t(p_{1:t}) \geq \tau \underbrace{\left(\frac{1}{t}\sum_{k=1}^t \mathbb{P}_{P_k}(\mathbf{u}_k > \lambda_k) - \mathbb{P}_{P_0}(\mathbf{u}_0 > \lambda_0, \mathbf{z}_0 \leq \tau)\right)}_{:=B_t}, \forall t \geq 1.$$

*Proof.* The proof technique is inspired by the derivation presented in Amoukou et al. [1]; see Eqs. (12)–(15) in their paper. To derive a lower bound on the true running test risk, we first apply Markov's inequality and then invoke Assumption 1:

$$\bar{R}_t(p_{1:t}) = \frac{1}{t}\sum_{k=1}^t \mathbb{E}_{P_k}[\mathbf{z}_k] \overset{\text{Markov's}}{\geq} \frac{1}{t}\sum_{k=1}^t \mathbb{P}_{P_k}(\mathbf{z}_k > \tau) \cdot \tau =$$

$$\tau\left(\frac{1}{t}\sum_{k=1}^t \mathbb{P}_{P_k}(\mathbf{u}_k > \lambda_k, \mathbf{z}_k > \tau) + \mathbb{P}_{P_k}(\mathbf{u}_k \leq \lambda_k, \mathbf{z}_k > \tau)\right) =$$

$$\tau\left(\frac{1}{t}\sum_{k=1}^t \mathbb{P}_{P_k}(\mathbf{u}_k > \lambda_k) - \mathbb{P}_{P_k}(\mathbf{u}_k > \lambda_k, \mathbf{z}_k \leq \tau) + \mathbb{P}_{P_k}(\mathbf{u}_k \leq \lambda_k, \mathbf{z}_k > \tau)\right) \overset{\text{Ass.1}}{\geq}$$

$$\tau\left(\frac{1}{t}\sum_{k=1}^t \mathbb{P}_{P_k}(\mathbf{u}_k > \lambda_k) - \mathbb{P}_{P_0}(\mathbf{u}_0 > \lambda_0, \mathbf{z}_0 \leq \tau)\right)$$

$\square$

If the risk definition includes conditioning on $\mathbf{x}_{1:k-1}$, i.e., $R_k(p_k) := \mathbb{E}_{P_k}[\mathbf{z}_k|\mathbf{x}_{1:k-1}]$, the proof proceeds analogously, with the only change being the use of conditional Markov's inequality. In this case, the resulting bound holds *almost surely*.

## B.3 PFA Control Guarantee

**Proposition 2.** *The unsupervised alarm $\Phi_t^b$ (Eq. 7) for the TTA sequential test (Eq. 5) satisfies a probability of false alarm (PFA) control guarantee:*

$$\mathbb{P}_{H_0}(\exists t \geq 1, \Phi_t^b = 1) \leq \alpha_{test} + \alpha_{source}.$$

*Proof.* The proof closely follows the PFA proof for the supervised alarm from Podkopaev and Ramdas [30], see Appendix D there. To show the PFA guarantee we proceed as:[4]

$$\mathbb{P}_{H_0}(\exists t \geq 1, \Phi_t^b = 1) = \mathbb{P}_{H_0}(\exists t \geq 1, L_t^b - U > \epsilon_{\text{tol}}) =$$
$$\mathbb{P}_{H_0}(\exists t \geq 1, (L_t^b - \bar{R}_t) - (U - R_0) > \epsilon_{\text{tol}} - (\bar{R}_t - R_0)) \leq$$
$$\mathbb{P}_{H_0}(\exists t \geq 1, (L_t^b - \bar{R}_t) - (U - R_0) > 0),$$

where the inequality follows from the fact that under $H_0$ (Eq. 5), we have that $\epsilon_{\text{tol}} \geq \bar{R}_t - R_0$. Since $\exists t \geq 1, (L_t^b - \bar{R}_t) - (U - R_0) > 0$ implies that either $\exists t \geq 1, L_t^b - \bar{R}_t > 0$ or $U - R_0 < 0$, we can use union bound to continue as:

$$\mathbb{P}_{H_0}(\exists t \geq 1, (L_t^b - \bar{R}_t) - (U - R_0) > 0) \leq$$
$$\mathbb{P}(\exists t \geq 1, L_t^b - \bar{R}_t > 0) + \mathbb{P}(U - R_0 < 0) \leq \alpha_{\text{test}} + \alpha_{\text{source}},$$

where the last inequality follows from the fact that $L_t^b$ is a lower bound confidence sequence for the lower bound $B_t$, i.e., $\mathbb{P}(B_t \geq L_t^b, \forall t \geq 1) \geq 1 - \alpha_{\text{test}}$, together with Proposition 1, which ensures $\bar{R}_t \geq B_t, \forall t \geq 1$, and the fact that $U$ is the upper bound of the confidence interval for $R_0$.

$\square$

### B.4  Tighter Bound for 0-1 Loss

**Corollary 1.** *For a $0$-$1$ loss function, assume that for a sequence of losses $\mathbf{z}_{0:t}$, a sequence of loss proxies $\mathbf{u}_{0:t}$ together with thresholds $\lambda_0, \ldots, \lambda_t \in \mathbb{R}$ satisfying Assumption 1 are available. Then the running test risk can be lower bounded as*

$$\bar{R}_t(p_{1:t}) \geq \frac{1}{t} \sum_{k=1}^{t} \mathbb{P}_{P_k}(\mathbf{u}_k > \lambda_k) - \mathbb{P}_{P_0}(\mathbf{u}_0 > \lambda_0, \mathbf{z}_0 = 0), \forall t \geq 1.$$

*Proof.* This (tighter) bound follows from the fact that for 0-1 loss, Markov's inequality is unnecessary due to the binary nature of the loss:

$$\bar{R}_t(p_{1:t}) = \frac{1}{t} \sum_{k=1}^{t} \mathbb{E}_{P_k}[\mathbf{z}_k] \stackrel{(0\text{-}1)}{=} \frac{1}{t} \sum_{k=1}^{t} \mathbb{P}_{P_k}[\mathbf{z}_k = 1].$$

The remainder of the proof then proceeds identically to that of Proposition 1. $\square$

Observe how the lower bound for 0-1 loss is the same as the lower bound for a general (bounded, non-negative) loss in Proposition 1 up to the loss threshold $\tau$. Additionally, we leave out the loss threshold $\tau$ from Assumption 1, i.e., we assume that the proxy sequence $\mathbf{u}_{0:t}$ is such that:

$$\frac{1}{t} \sum_{k=1}^{t} \mathbb{P}_{P_k}(\mathbf{u}_k > \lambda_k, \mathbf{z}_k = 0) \leq \mathbb{P}_{P_0}(\mathbf{u}_0 > \lambda_0, \mathbf{z}_0 = 0) + \frac{1}{t} \sum_{k=1}^{t} \mathbb{P}_{P_k}(\mathbf{u}_k \leq \lambda_k, \mathbf{z}_k = 1).$$

### B.5  PFA for "Probability of High Loss" Test

**Proposition 3.** *The unsupervised alarm $\Phi_t^\tau$ (Eq. 8) for the 'probability of high loss' TTA sequential test (Eq. 9) satisfies a PFA control guarantee:*

$$\mathbb{P}_{H_0}(\exists t \geq 1, \Phi_t^\tau = 1) \leq \alpha_{test} + \alpha_{source}.$$

*Proof.* To simplify notation, denote with $\bar{R}_t^{\mathbb{P}} := \frac{1}{t} \sum_{k=1}^{t} \mathbb{P}_{P_k}(\mathbf{z}_k > \tau)$ and $R_0^{\mathbb{P}} := \mathbb{P}_{P_0}(\mathbf{z}_0 > \tau)$. From the proof of Proposition 1, it follows that $\bar{R}_t^{\mathbb{P}} \geq \frac{1}{\tau} B_t$, which, combined with the fact that $L_t^b$ is a lower bound confidence sequence for $B_t$, implies that $\mathbb{P}(\bar{R}_t^{\mathbb{P}} \geq \frac{1}{\tau} L_t^b, \forall t \geq 1) \geq 1 - \alpha_{\text{test}}$. Similarly,

---

[4]To simplify notation, we omit all arguments of the relevant risks and confidence sequences in the proof (e.g., we abbreviate $U(\mathbf{z}_0)$ as $U$).

since $U^b$ is an upper bound of the confidence interval for $\tau R_0^{\mathbb{P}}$, it follows that $\mathbb{P}(\frac{1}{\tau}U^b \geq R_0^{\mathbb{P}}) \geq 1 - \alpha_{\text{source}}$. The rest of the proof is then indentical to the proof of Proposition 2:

$$\mathbb{P}_{H_0}\left(\exists t \geq 1, \Phi_t^\tau = 1\right) = \mathbb{P}_{H_0}\left(\exists t \geq 1, \frac{1}{\tau}L_t^b - \frac{1}{\tau}U^b > \tilde{\epsilon}_{\text{tol}}\right) =$$

$$\mathbb{P}_{H_0}\left(\exists t \geq 1, (\frac{1}{\tau}L_t^b - \bar{R}_t^{\mathbb{P}}) - (\frac{1}{\tau}U - R_0^{\mathbb{P}}) > \tilde{\epsilon}_{\text{tol}} - (\bar{R}_t^{\mathbb{P}} - R_0^{\mathbb{P}})\right) \overset{H_0}{\leq}$$

$$\mathbb{P}_{H_0}\left(\exists t \geq 1, (\frac{1}{\tau}L_t^b - \bar{R}_t^{\mathbb{P}}) - (\frac{1}{\tau}U - R_0^{\mathbb{P}}) > 0\right) \leq$$

$$\mathbb{P}\left(\exists t \geq 1, \frac{1}{\tau}L_t^b - \bar{R}_t^{\mathbb{P}} > 0\right) + \mathbb{P}\left(\frac{1}{\tau}U - R_0^{\mathbb{P}} < 0\right) \leq \alpha_{\text{test}} + \alpha_{\text{source}} \ .$$

$\square$

## C  Algorithms

---

**Algorithm 1:** TTA with Risk Monitoring

---

**Input** : Calibration data $\mathcal{D}_{\text{cal}} = \{(\boldsymbol{x}_i, y_i)\}_{i=1}^{N_{\text{cal}}}$ with $(\boldsymbol{x}_i, y_i) \sim P_0$, test data $\mathcal{D}_{\boldsymbol{x}}^k = \{\boldsymbol{x}_i\}_{i=1}^{N_k}$ with $\boldsymbol{x}_i \sim P_k$, loss function $\ell$, proxy function $g$, source model $p_0$, tolerance level $\epsilon_{\text{tol}}$, significance levels $\alpha_{\text{source}}$ and $\alpha_{\text{test}}$, TTA method $h : (p_{k-1} \times \mathcal{D}_{\boldsymbol{x}}^k) \mapsto p_k$

1   Compute source losses $z_{0,i} = \ell(p_0(\boldsymbol{x}_i), y_i)$
2   Compute source loss proxies $u_{0,i} = g(\boldsymbol{x}_i, p_0)$
3   Find source thresholds $\hat{\lambda}_0, \hat{\tau} := \arg\max_{\lambda, \tau} \text{F1}(\lambda, \tau; \{(z_{0,i}, u_{0,i})\}_{i=1}^{N_{\text{cal}}})$
4   Compute upper bound $\hat{U}$ using $\{z_{0,i}\}_{i=1}^{N_{\text{cal}}}$ and $\alpha_{\text{source}}$
5   **for** $k \geq 1$ **do**
6      Perform TTA update $p_k = h(p_{k-1}, \mathcal{D}_{\boldsymbol{x}}^k)$
7      Compute losses of model $p_k$ on $\mathcal{D}_{\text{cal}}$: $z_{0,i}^{(k)} = \ell(p_k(\boldsymbol{x}_i), y_i)$
8      Compute loss proxies of model $p_k$ on $\mathcal{D}_{\text{cal}}$: $u_{0,i}^{(k)} = g(\boldsymbol{x}_i, p_k)$
9      Update proxy threshold $\hat{\lambda}_k := \arg\max_{\lambda} \text{F1}(\lambda, \hat{\tau}; \{(z_{0,i}^{(k)}, u_{0,i}^{(k)})\}_{i=1}^{N_{\text{cal}}})$
10     Compute loss proxies of model $p_k$ on $\mathcal{D}_{\boldsymbol{x}}^k$: $u_{k,i} = g(\boldsymbol{x}_i, p_k)$
11     Compute lower bound $\hat{L}_k^b$ using $\{u_{1,i}\}_{i=1}^{N_1}, \ldots, \{u_{k,i}\}_{i=1}^{N_k}, \{z_{0,i}\}_{i=1}^{N_{\text{cal}}}, \hat{\lambda}_{0:k}, \hat{\tau}, \alpha_{\text{test}}$
12     Compute alarm $\hat{\Phi}_k^b = \mathbb{1}\left[\hat{L}_k^b > \hat{U} + \epsilon_{\text{tol}}\right]$
13     **if** $\hat{\Phi}_k^b = 1$ **then**
14       Terminate TTA
15       **break**
16     **else**
17       Predict using $p_k$ on $\mathcal{D}_{\boldsymbol{x}}^k$: $\hat{y}_i = \arg\max_c p_k(\boldsymbol{x}_i)_c$
18       **continue**

---

**Algorithm 2:** Online Threshold Calibration

---

**Input** : Calibration data $\mathcal{D}_{\text{cal}} = \{(\boldsymbol{x}_i, y_i)\}_{i=1}^{N_{\text{cal}}}$ with $(\boldsymbol{x}_i, y_i) \sim P_0$, loss function $\ell$, proxy function $g$, source model $p_0$, TTA models $p_1, \ldots, p_t$
**Output** : loss threshold $\hat{\tau}$, proxy thresholds $\hat{\lambda}_{0:t}$

1   Compute source losses $z_{0,i} = \ell(p_0(\boldsymbol{x}_i), y_i)$
2   Compute source loss proxies $u_{0,i} = g(\boldsymbol{x}_i, p_0)$
3   Find source thresholds $\hat{\lambda}_0, \hat{\tau} := \arg\max_{\lambda, \tau} \text{F1}(\lambda, \tau; \{(z_{0,i}, u_{0,i})\}_{i=1}^{N_{\text{cal}}})$
4   **for** $k = 1 \to t$ **do**
5      Compute losses of model $p_k$ on $\mathcal{D}_{\text{cal}}$: $z_{0,i}^{(k)} = \ell(p_k(\boldsymbol{x}_i), y_i)$
6      Compute loss proxies of model $p_k$ on $\mathcal{D}_{\text{cal}}$: $u_{0,i}^{(k)} = g(\boldsymbol{x}_i, p_k)$
7      Update proxy threshold $\hat{\lambda}_k := \arg\max_{\lambda} \text{F1}(\lambda, \hat{\tau}; \{(z_{0,i}^{(k)}, u_{0,i}^{(k)})\}_{i=1}^{N_{\text{cal}}})$
8   **return** $\hat{\tau}, \hat{\lambda}_{0:t}$

# D   Experimental Details

## D.1   Datasets

- **ImageNet-C** [12]: This dataset applies 15 types of algorithmic corruptions (e.g., Gaussian noise, blur, weather effects, digital distortions) at five severity levels to the original ImageNet [60] validation set. The dataset preserves the original 1,000-class classification task, using the same labels and image resolutions. In our setup, we focus on Gaussian noise corruption.
- **Yearbook** [8]: This dataset contains portraits of American high school students taken over eight decades, capturing changes in visual appearance due to evolving beauty standards, cultural norms, and demographics. We follow the Wild-Time preprocessing and evaluation protocol [45], resulting in 33,431 grayscale images (32×32 pixels) labeled with binary gender. Images from 1930–1969 are used for training, and those from 1970–2013 for testing.
- **FMoW-Time**: The Functional Map of the World (FMoW) dataset [61] consists of 224×224 RGB satellite images categorized into 62 land-use classes. Distribution shift arises from technological and economic changes that alter land usage over time. FMoW-Time [45] is a temporal split of FMoW-WILDS [61, 62], dividing 141,696 images into a training period (2002–2014) and a testing period (2015–2017).

## D.2   TTA Methods

We evaluate our monitoring tool across several TTA methods, which differ in the set of adapted parameters (e.g., normalization layers, full model, classification head) and in their objective functions (e.g., entropy minimization, information maximization, log-likelihood maximization):

- **TENT**[42] updates normalization layers by minimizing test entropy.
- **SHOT** [19] adapts normalization layers using information maximization and self-supervised pseudo-labeling to align target representations with a frozen source classifier.
- **SAR** [27] updates normalization layers via an entropy minimization objective. It filters out high-entropy samples and guides adaptation toward flatter minima.
- **CoTTA** [41] updates all model parameters using a student-teacher approach on augmentation averaged predictions. It also employs stochastic weight restoration to mitigate forgetting.
- **T3A** [15] adjusts only the final linear classifier by computing class-wise pseudo-prototypes from confident, normalized representations.
- **STAD** [35] updates only the last linear layer by tracking the evolution of feature representations with a probabilistic state-space model.

## D.3   Implementation Details

All experiments are performed on NVIDIA RTX 6000 Ada with 48GB memory. We plot the mean and standard deviations over 20 runs. The variability across runs stems from calibration set sampling and test sample shuffling with different random seeds. For Fig. 5 and **??**, we use 10 random seeds.

For each TTA method, we use the default hyperparameters proposed in the respective paper. We use a test batch size of 32 for ImageNet and 64 for Yearbook and FMoW-Time.

We use the `confseq` package [63] by [13] to compute the conjugate-mixture empirical Bernstein confidence lower bound on the target risk. This confidence sequence framework supports tuning for an intrinsic time $t_{\mathrm{opt}}$, which we set by default to the first 25% of the sequence length for all experiments. To implement the baseline $\hat{L}_t^d$ from Amoukou et al. [1], we use the same loss proxy—uncertainty—as in our method.

# E  Further Related Work

**Error and accuracy estimation**   aims to assess model performance on unlabeled test data, which is often subject to distribution shift [64, 65]. This is typically achieved via model uncertainty [66, 4, 67–69] or model disagreement [70, 71, 16, 33, 18, 72, 73]. Uncertainty-based methods exploit the predictive distribution of the model— for example through the maximum class probability [66] or the true class probability [4]—and learn a threshold to distinguish correctly from incorrectly predicted samples [67–69]. Disagreement-based error prediction methods leverage the theoretical equivalence between model disagreement and test error under calibration [70, 16]. However, these methods often require training multiple models—sometimes even from different architectures [71, 70]. Closest to our work [18, 31, 74], estimate the accuracy of TTA methods based on disagreement. Notably, by exploiting theoretical results from [70], Lee et al. [18] proposes an accuracy estimation method based on dropout disagreement. They differ from our work by (i) providing an estimator of test risk directly while we are interested in signaling a significant increase in test risk compared to the source risk; as such (ii) their method does not come with guarantees on the false alarm rate; and (iii) they require calibration (their Definition 3.3) to preserve theoretical validity of their risk estimator while we rely on separability of high and low error samples (Assumption 1).

**TTA robustness**   Recent work has identified several scenarios where TTA methods tend to degrade. These include adaptation under non-stationary test distributions [41, 75, 46, 54, 6, 22, 76, 35], label shift [10, 26, 3, 46], mixed domains within a test batch [27, 54, 22], small test batch sizes [27, 54, 22, 6, 35], and adaptation in the presence of malicious samples [29, 43]. Most such work on TTA robustness has focused on proposing more robust TTA methods and developing evaluation benchmarks [10, 22, 46, 76]. Liu et al. [77] analyze failure cases of the related test-time training paradigm, which requires a self-supervised objective during training. They derive an upper bound on test risk dependent on the effectiveness of the self-supervised loss. In contrast, our approach does not require any modification to the training procedure and provides guarantees that hold regardless of the model's original training objective. Also related to our work is research on TTA model collapse—where models degenerate to trivial solutions during adaptation [31, 27, 18, 78]—and efforts to identify optimal reset mechanisms that revert the model back to its source parameters during deployment [26, 18, 78]. In contrast, we propose a general-purpose monitoring tool that provides statistical guarantees on risk control for arbitrary TTA methods. Rather than focusing on a specific mitigation strategy, our tool can inform a range of interventions—such as resetting the model to its source for continued adaptation or taking it offline entirely for retraining [79].

# F  Impact Statement

This work introduces a statistically grounded framework for detecting risk violations during TTA, a key challenge for deploying machine learning models in dynamic, real-world environments. By enabling risk monitoring without access to labels, our approach promotes safer and more trustworthy use of TTA methods—particularly in high-stakes domains such as healthcare, autonomous systems, and finance, where undetected model failure can have serious consequences. Our method complements existing adaptation techniques by offering a safeguard against silent performance degradation and model collapse, helping practitioners determine when adaptation is no longer effective. In doing so, it supports more responsible and robust deployment of adaptive models. While the framework provides high-probability guarantees, misuse or overreliance could lead to overconfidence in model reliability. We therefore emphasize the importance of understanding its assumptions and limitations. Overall, this work contributes to the safe deployment of adaptive machine learning models under distribution shift.

