# OpenReview forum: "Monitoring Risks in Test-Time Adaptation"
_NeurIPS.cc/2025/Conference — NeurIPS 2025 poster_

### Official Review · Reviewer_LLXY · 2025-06-17

**Clarity:** 4
**Significance:** 2
**Originality:** 3
**Rating:** 4
**Confidence:** 2

**Summary:**

This paper presents a novel unsupervised risk monitoring framework for test-time adaptation (TTA), addressing the critical challenge of detecting performance degradation in models undergoing continuous online adaptation. The authors systematically extend sequential testing frameworks to dynamic TTA scenarios, leveraging confidence sequences and loss proxies (e.g., model uncertainty) to enable rigorous risk tracking without labeled test data. The research is theoretically sound, experimentally comprehensive, and practically relevant, particularly for safety-critical application.

**Questions:**

I still hope that you can place more emphasis on the problem of how to select proxies based on real-world scenarios. Comparing the impacts of different proxy choices across a wider range of scenarios would likely contribute more than the current paper does.

**Ethical Concerns:**

["NO or VERY MINOR ethics concerns only"]

**Final Justification:**

Thank you for the authors’ response. I found some experimental improvements in this reply and in the responses to other reviewers, which align with my suggestion to explore richer proxy choices based on real-world scenarios. This has positively influenced my overall assessment of the paper.

Regarding my concern that the technical contributions of the paper are limited, the authors acknowledge that the methods they actually use are ones that are already widely adopted in the field. Their response repeatedly emphasizes the contribution of bridging two research areas. However, I never denied this aspect, and my original score was given with this contribution already taken into account.

Therefore, I will consider raising my score specifically in light of the broader set of proxy choices explored. However, this alone is not sufficient for me to raise my score from a 3 to a 4. After reviewing all the other reviewers’ comments, I feel even more confident that this is a borderline submission. I would have preferred to give a 3.5 if that were an option.

To encourage the authors to further explore richer and more novel technical contributions, I am increasing my score to a 4, while lowering my confidence accordingly.

**Limitations:**

Yes.

**Paper Formatting Concerns:**

No formatting concerns.

**Quality:**

4

**Strengths And Weaknesses:**

Strengths:

1. I appreciate the theoretical rigor of this paper, which provides solid theoretical foundations for risk management in TTA.

2. I also admire the writing style, which is remarkably clear and reader-friendly.

3. I appreciate the sufficiently rigorous practice-based discussions in both the main text and the appendix.

Weaknesses:

1. I mainly worry that the practical value of this paper may be insufficient, as using proxy losses to judge model performance in unsupervised scenarios is already common knowledge, and I believe the contribution of proposing this approach might be very limited. In reality, the bigger challenge is that practitioners often do not know how to select appropriate proxies for specific scenarios, yet the paper provides minimal discussion on this critical aspect.

2. My impression is that while the paper approaches the topic from a different perspective than previous work, there is little innovation in specific techniques. It still leverages model uncertainty (with a limited comparison to another metric) and sets thresholds for judgment—approaches that have become well-established in the field.

---

> ### Author Rebuttal · Authors · 2025-07-30
>
> We thank you for your feedback and reviewing efforts. We address your points below.
>
> > *In reality, the bigger challenge is that practitioners often do not know how to select appropriate proxies for specific scenarios, yet the paper provides minimal discussion on this critical aspect.*
>
> > *I still hope that you can place more emphasis on the problem of how to select proxies based on real-world scenarios. Comparing the impacts of different proxy choices across a wider range of scenarios would likely contribute more than the current paper does.*
>
> We agree that the selection of appropriate loss proxy is crucial for the effectiveness of the monitoring tool. However, we are somewhat puzzled by the statement that “*the paper provides minimal discussion on this critical aspect.”* as we have dedicated a large part of our paper to this very topic. Specifically, in Section 3.3 we propose a concrete loss proxy based on model uncertainty (as well as a concrete online threshold calibration procedure in Section 3.4), and then later show in Sections 5.1 and 5.2 that our choices lead to effective monitoring in real TTA scenarios across five TTA methods and four types of distribution shifts. Moreover, in Section 5.4, we explain why the loss proxy should be chosen with the characteristics of the given TTA method in mind, by showing that model uncertainty is unsuitable for last-layer TTA methods due to the normalization of classification heads. To address this, we propose a different proxy based on distances to class prototypes, which proves effective for last-layer methods. Additionally, in Appendix A.3 we report results when using the energy score [1] as loss proxy, showing that it underperforms compared to our default choice of model uncertainty. During the rebuttal, we have also ran an additional ablation of using entropy as a loss proxy (see our response to d5NY) and will add it to our loss proxy ablation in Appendix A.3 in the camera-ready version.
>
> If you still feel there is something we should discuss more when it comes to selecting the loss proxy for TTA monitoring, or if you have some concrete real-world scenarios in mind (besides the experiments reported in the paper), please let us know and we will be happy to provide further details or run additional experiments.
>
> > *I mainly worry that the practical value of this paper may be insufficient, as using proxy losses to judge model performance in unsupervised scenarios is already common knowledge, and I believe the contribution of proposing this approach might be very limited.*
>
> > *It still leverages model uncertainty (with a limited comparison to another metric) and sets thresholds for judgment—approaches that have become well-established in the field.*
>
> We would like to clarify that our contribution goes beyond simply using raw loss proxies as a heuristic (e.g., model uncertainty) to “*judge model performance in an unsupervised scenario.*” **Rather, our main contribution is to extend the statistical framework of sequential risk monitoring [2] to the TTA setting**, which additionally requires figuring out the necessary assumptions to preserve theoretical guarantees (Section 3.2) and tackling the challenge of online calibration to ensure that loss proxy remains informative under data shifts and model adaptation (Section 3.4). This allows us to track test performance of TTA models without labels and without assumptions on the nature of the distribution shift. To the best of our knowledge, no existing work provides such an unsupervised monitoring tool tailored for TTA. If you have some concrete prior work in mind that we might have overlooked, we would appreciate if you could point it out to us.
>
> We would also add that the importance/practical value of our work has been recognized by other reviewers. For example, reviewer d5NY noted that "*this paper addresses a problem of practical importance in TTA*" and “*shows good potential to address real-world deployment challenges*”, while reviewer PuqZ stated that "*the paper fills a significant gap by risk monitoring for TTA without test labels.*"
>
> Since even you yourself wrote “*This paper presents a novel unsupervised risk monitoring framework for test-time adaptation (TTA), addressing the critical challenge of detecting performance degradation in models undergoing continuous online adaptation.”,* we are left wondering what exactly you mean when you later state *“that the practical value of this paper may be insufficient”*? If you could elaborate more on your concrete worries about the lack of practical value of our approach, we would be very happy to engage in further discussions.
>
> > *My impression is that while the paper approaches the topic from a different perspective than previous work, there is little innovation in specific techniques.*
>
> As mentioned earlier, to the best of our knowledge, our work is the first to extend risk monitoring tools [2] to the TTA setting. **We believe that bridging the gap between the test-time adaptation (TTA) and risk monitoring literatures is a valuable and timely contribution, as it enables the safer deployment of TTA methods in real-world applications.** To date, performance monitoring has largely been absent in TTA or has relied on heuristic measures. Our work addresses precisely this gap by introducing a more principled and rigorous monitoring approach.
>
> If there are specific prior works you believe we have overlooked, we would greatly appreciate if you could share concrete references. As we discuss in the paper, the most relevant prior work in our opinion is [3]. However, we believe our contributions represent significant advances beyond the framework proposed in [3]. Notably [3] focuses on static models, whereas our work tackles the more complex and practically relevant setting of adapted models (TTA). We additionally introduce several theoretical extensions, the most important of which is arguably the support for monitoring the classification error as a risk measure which is the most standard performance measure in TTA applications (please refer to Appendix A.1 for a more detailed discussion on the differences of our work compared to [3]).
>
> If you feel that we have addressed your concerns, we would appreciate it if you would consider updating your score.
>
> [1] Liu et al. Energy-based out-of-distribution detection. NeurIPS 2020
>
> [2] Podkopaev et al. Tracking the risk of a deployed model and detecting harmful distribution shifts. ICLR 2022
>
> [3] Amoukou et al. Sequential Harmful Shift Detection Without Labels. NeurIPS 2024

---

> > ### Comment · Reviewer_LLXY · 2025-08-06
> >
> > Thanks for the authors’ response. I found some experimental improvements in this reply and in the responses to other reviewers, which align with my suggestion to explore richer proxy choices based on real-world scenarios. This has positively influenced my overall assessment of the paper.
> >
> > Regarding my concern that the technical contributions of the paper are limited, the authors acknowledge that the methods they actually use are ones that are already widely adopted in the field. Their response repeatedly emphasizes the contribution of bridging two research areas. However, I never denied this aspect, and my original score was given with this contribution already taken into account.
> >
> > Therefore, I will consider raising my score specifically in light of the broader set of proxy choices explored. However, this alone is not sufficient for me to raise my score from a 3 to a 4. After reviewing all the other reviewers’ comments, I feel even more confident that this is a borderline submission. I would have preferred to give a 3.5 if that were an option.
> >
> > To encourage the authors to further explore richer and more novel technical contributions, I am increasing my score to a 4, while lowering my confidence accordingly.

---

> ### Author Response · Authors · 2025-08-05
>
> Dear Reviewer LLXY,
>
> we thank you again for your reviewing effort. Since the discussion period is slowly approaching the end, we were wondering if you had a chance to look at our rebuttal?
>
> We remain available for any potential follow-up questions.
>
> Kind regards,
>
> Authors

---

> ### Author Response · Authors · 2025-08-06
>
> Dear Reviewer LLXY,
>
> Thanks for your continued engagement and for increasing your score. We are glad we were able to resolve your concerns regarding the choice of the loss proxy.
>
> We would like to politely push back on your interpretation that “*the authors acknowledge that the methods they actually use are ones that are already widely adopted in the field*”. We did not make this claim in the rebuttal and maintain that our work is novel, technically non-trivial, and a timely contribution to the TTA field.
>
> Since it seems that your concerns regarding the novelty of our work persist to some extent, we would highly appreciate if you could point out any concrete references that you have in mind and that are the reasons behind these novelty concerns. Having concrete references would be very helpful in allowing us to more clearly highlight our novel technical contributions.
>
> Kind regards,
>
> Authors

---

### Official Review · Reviewer_d5NY · 2025-06-22

**Clarity:** 3
**Significance:** 3
**Originality:** 3
**Rating:** 4
**Confidence:** 3

**Summary:**

This paper proposes a method for monitoring risks in test-time adaptation (TTA). Monitoring risk has been done before. But previous work usually only consider risk monitoring for fixed models after they are deployment. This paper is the first one (that I know of) that deals with the risk monitoring in the test-time adaptation (TTA), where the model can adapt itself after deployment. This paper proposes an algorithm based some theoretical results motivated by previous, especially [1] and [30]

**Questions:**

Please refer to "weakness" above

**Ethical Concerns:**

["NO or VERY MINOR ethics concerns only"]

**Final Justification:**

The rebuttal addressed most of my questions. I will maintain the positive rating.

**Limitations:**

yes

**Quality:**

3

**Strengths And Weaknesses:**

Strength
---------

Test-time adaptation has become popular recently. Most TTA work focuses on developing new TTA algorithms that achieve higher accuracy. There is little work on studying the problem of detecting when the risk monitoring (i.e. the performance of the model degrades too much after adaptation for a long time, so the model needs to be retrained). I think this paper addresses a problem of practical importance in TTA.

The proposed method seems to be sound (although I am not an expert in theoretical machine learning, so I cannot comment on how novel or significant those theoretical results are). Apart from theoretical results, this paper also provides a concrete method that can be operationalized. The method is quite simple to implement in practice.

Weakness
------------

1) From what I can understand, the key idea of the proposed method is to define a loss proxy u that does not require ground-truth labels. This paper chose a particular proxy in Sec 3.3 using some kind of uncertainty. But it is not clear to me whether this particular choice is somehow connected to the theoretical results in previous sections. There are many other ways of defining uncertainty (e.g. the entropy used in TENT). Would the method still work if you use other uncertainty metric? Is the particular uncertainty metric used in this paper better than other alternatives?

2) For most TTA methods (e.g. TENT), they simply optimize some unlabeled loss function during adaptation. To me, the uncertainty measure in Sec 3.3 can also be used as the loss function in TTA. In other words, the TTA algorithm can directly try to optimize this risk during testing. Intuitively, the TTA method will "overfit" to this risk measure. I am interested in whether the proposed method still works in this case.

3) The experiment compares different methods by showing some curves. I think it is important to define some numerical metric based on those curves (e.g. in other fields, people use AUC based on precision/recall curve). Otherwise, it is impossible for any follow-up work to directly compare with this paper.

---

> ### Author Rebuttal · Authors · 2025-07-30
>
> We thank you for your reviewing efforts and feedback, it is much appreciated. We address your questions below.
>
> > *From what I can understand, the key idea of the proposed method is to define a loss proxy u that does not require ground-truth labels.*
>
> While coming up with a suitable loss proxy that doesn’t require labels is an important part of our method (as done in Section 3.3), we would like to clarify that it is not by itself our main contribution/idea. Instead our main focus is to extend the risk monitoring based on sequential testing to the setting of TTA, which additionally requires figuring out the necessary assumptions to preserve theoretical guarantees (Section 3.2) and tackling the challenge of online calibration to ensure that loss proxy remains informative under data shifts and model adaptation (Section 3.4).
>
> > *This paper chose a particular proxy in Sec 3.3 using some kind of uncertainty. But it is not clear to me whether this particular choice is somehow connected to the theoretical results in previous sections.*
>
> According to our theoretical results in Section 3.2, any loss proxy can be used within our risk monitoring framework, provided it satisfies Assumption 1 (which ensures that the lower bound on the test error from Proposition 1 remains valid). Therefore, our proposal to use model uncertainty in Section 3.3 is motivated by our empirical observation (Section 5.2) that it consistently yields valid and tight lower bounds across a diverse set of TTA methods and shift types. Moreover, unlike some previously proposed proxies (e.g., those based on separately fitted error estimators [1]), model uncertainty is particularly appealing due to its simplicity—it requires no additional model parameters or extra forward passes on the test data. In addition to these empirical motivations, we also present a small theoretical justification in Section 3.3, arguing that model uncertainty approximates the true conditional risk up to the calibration error.
>
> We would like to thank you for raising this point. In the camera-ready, we will make sure to better explain the connection between the theory for general loss proxies in Section 3.2 and our concrete choice of model uncertainty in Section 3.3.
>
> > *There are many other ways of defining uncertainty (e.g. the entropy used in TENT). Would the method still work if you use other uncertainty metric? Is the particular uncertainty metric used in this paper better than other alternatives?*
>
> As explained in our previous response, our monitoring method is compatible with a range of different loss proxies. However, in our experiments, we found that the proposed model uncertainty proxy (Section 3.3) leads to more effective detection than several other popular uncertainty measures. Specifically, in Appendix A.3, we compare model uncertainty against the energy score [2], a widely used OOD detection metric. As shown in Figure 7, the energy score underperforms relative to model uncertainty, yielding looser lower bounds on test error and consequently leading to greater detection delays.
>
> Additionally, in Section 5.4 we further exemplify the flexibility of our framework to support various loss proxies, by explaining why model uncertainty is unsuitable for last-layer TTA methods  (due to the normalization applied to the classification head). In such cases, we propose using the distance to the closest class prototype as an alternative loss proxy.  Our findings in Section 5.4 highlight the importance of aligning the choice of loss proxy with the characteristics of the specific TTA approach.
>
> **Based on your suggestion we have also run the experiment with using entropy $g(\textbf{x}, p) = - \sum_{c=1}^C p(\textbf{x})_c \log p(\textbf{x})_c$  instead of model uncertainty as a loss proxy.** The results—summarized in the tables below due to our inability to attach plots—show that entropy yields detection delays that are comparable to, or slightly worse than, those from model uncertainty, depending on the shift type. We will include these entropy results in our loss proxy ablation study in Appendix A.3 and would like to thank you for the suggestion.
>
> ### Average Detection Delay ($\downarrow$) on ImageNet-C Severity 0→5:
> |  | SOURCE | TENT | SAR | SHOT | CoTTA |
> | --- | --- | --- | --- | --- | --- |
> | Entropy | 371 | 361 | 456  | 302 | 311 |
> | Uncertainty | **297** | **269** | **424** | **237** | **241** |
>
>
> ### Average Detection Delay ($\downarrow$) on ImageNet-C Severity 5:
>
> |  | SOURCE | TENT | SAR | SHOT | CoTTA |
> | --- | --- | --- | --- | --- | --- |
> | Entropy | 9 | 13 | 15 | 13 | 7 |
> | Uncertainty | 10 | 14 | 17 | 13 | 7 |
>
> > *For most TTA methods (e.g. TENT), they simply optimize some unlabeled loss function during adaptation. To me, the uncertainty measure in Sec 3.3 can also be used as the loss function in TTA. In other words, the TTA algorithm can directly try to optimize this risk during testing. Intuitively, the TTA method will "overfit" to this risk measure. I am interested in whether the proposed method still works in this case.*
>
> Thank you for raising this subtle point. **In short, yes—our risk monitoring remains valid and effective even when the same uncertainty measure is used both as the objective function in TTA adaptation and as a loss proxy in our monitoring tool, as demonstrated by the TENT results when entropy is used as a proxy** (see the tables in the previous answer)**.** This is because we do not use the raw loss proxy values directly in our alarm function. Doing so could indeed risk the type of “overfitting” you mention, where entropy minimization (as in TENT) reduces the proxy values on average, potentially causing the alarm to fail due to “overconfidence”.
>
> Instead, our alarm function relies on counting the number of test points whose loss proxy exceeds a carefully calibrated threshold (see Proposition 1). Therefore, if the loss proxy values decrease due to adaptation (e.g., entropy minimization), this effect is compensated by adjusting the threshold accordingly through our online calibration procedure (see Section 3.4), thus preserving the reliability of the alarm mechanism. Such adaptive thresholding is also the reason why our monitoring remains effective even when “overfitting” in TENT leads to the model collapse, as we show in Section 5.3.
>
> > *The experiment compares different methods by showing some curves. I think it is important to define some numerical metric based on those curves (e.g. in other fields, people use AUC based on precision/recall curve). Otherwise, it is impossible for any follow-up work to directly compare with this paper.*
>
> In our experiments, we aimed to follow the experimental setup used in prior work on risk monitoring based on sequential testing [1, 3]. We also believe that presenting the results in their current visual format helps with better understanding of our method. That said, we agree that including quantitative metrics would be valuable for future benchmarking. As in the tables above, we will summarize the results using the average detection delay—defined as the difference between the time at which our unsupervised lower bound crosses the performance threshold $\hat{t}\_{\text{min}}^b$ and the time at which the “true” test risk does so $\hat{t}\_{\text{min}}^a$—in the camera-ready version. Additionally, we will report the average delta associated with Assumption 1  (as defined in line 284 of the manuscript), as this metric directly reflects both the validity and tightness of our estimated lower bounds on the test error. If you have any further suggestions in this regard, we would be happy to incorporate them.
>
> We thank you again for your efforts in reviewing our work. If you feel like we have sufficiently addressed your concerns, we would appreciate you raising your score.
>
> [1] Amoukou et al. Sequential Harmful Shift Detection Without Labels. NeurIPS 2024
>
> [2] Liu et al. Energy-based out-of-distribution detection. NeurIPS 2020
>
> [3] Podkopaev et al. Tracking the risk of a deployed model and detecting harmful distribution shifts. ICLR 2022

---

> > ### Comment · Reviewer_d5NY · 2025-08-05
> >
> > The rebuttal addressed most of my questions. I will maintain the positive rating.

---

> > > ### Author Response · Authors · 2025-08-07
> > >
> > > Dear Reviewer d5NY,
> > >
> > > thanks for your reply and for your overall positive assessment of our paper. We remain available in case of any additional follow-up questions.
> > >
> > > Best,
> > >
> > > Authors

---

### Official Review · Reviewer_PuqZ · 2025-07-01

**Clarity:** 3
**Significance:** 3
**Originality:** 3
**Rating:** 4
**Confidence:** 4

**Summary:**

This paper proposes a risk monitoring method for TTA, aiming to address the challenge of detecting potential performance degradation during TTA when encountering distribution shifts. It extends sequential testing to the unsupervised TTA setting with no test labels used. Key components include a loss proxy using model uncertainty, online threshold calibration with labeled source subset, and theoretical analyses for false alarm control. It is validated across datasets and TTA methods, showing its effectiveness in detecting risk violations to prevent TTA collapse.

**Questions:**

1. How does the framework perform when Assumption 1 is persistently violated? Is there any failure case?
2. Can the framework be tuned to prioritize early detection over false alarm rate control for safety-critical applications? How to deal with the trade-off.

**Ethical Concerns:**

["NO or VERY MINOR ethics concerns only"]

**Final Justification:**

I have read the authors' detailed response. It has successfully addressed several of my concerns, particularly regarding the framework's performance in dynamic environments and the overhead of the monitoring tool. The new experimental results on the continual ImageNet-C setting are a valuable addition that I believe should be included in the final version. While I still harbor some concerns regarding Assumption 1 and the reliance on labeled data, as these factors may restrict the real-world applicability. Given the improved clarity, I intend to maintain my borderline accept rating while increasing my confidence in this assessment.

**Limitations:**

1. Detection Delay: The approach introduces inevitable delay, which may may impact time-sensitive applications.
2. Label Requirement: Verifying the key assumption (Assumption 1) needs labeled data.  Additionally, the requirements for D_cal from the source domain may violate the source-free principle to some extent.

**Quality:**

3

**Strengths And Weaknesses:**

## Strengths
1. **Novel Unsupervised Monitoring Approach**: TTA methods are proved to be vulnerable to some environments or shifts, knowing when TTA methods collapse or worse than source model is a critical task. The paper fills a significant gap by risk monitoring for TTA without test labels, leveraging model uncertainty as a proxy and extending sequential testing frameworks.
2. **Theoretical Rigor**: The approach is grounded in confidence sequences, providing time-uniform guarantees, the authors derive false alarm control and validate assumptions both theoretically and empirically.
3. **Practical Adaptability**: Online threshold calibration and alternative proxies enhance robustness to different TTA methods and different shifts without specific designs, which shows good potential to address real-world deployment challenges.

## Weaknesses
1. **Dynamic Environment Assumptions**: the framework assumes that the test domain is under stationary distributions. While more pratically it may not hold in scenarios with continual distribution shifts. In this more challenging setting, can risk monitoring also work well? It should handle the rapid distribution shift when facing domain change. Some theoretical anlayses and extensive experiments with continual TTA on the whole ImageNet-C with the CoTTA proposed setting may better prove its effectiveness.
2. **Limited Collapse Prevention Comparisons**: While the framework detects collapse, It lacks of comparison with other collapse prevent methods. For example, some baseline methods like SAR’s resetting strategy, using $D_{cal}$ as a validation set for comparison between current model and source model.
3. **Unquantified Computational Overhead**: The paper omits metrics (e.g., FLOPS, inference time) to characterize the monitoring framework’s computational cost, which is critical for resource-constrained deployments.

---

> ### Author Rebuttal · Authors · 2025-07-30
>
> We thank you for your reviewing efforts, it is much appreciated. We address your questions below.
>
> > Dynamic Environment Assumptions: the framework assumes that the test domain is under stationary distributions. [...]
>
> We would like to clarify that we do not assume that "*test domain is under stationary distribution*". Instead, as we write in Section 2 (lines 55-57):
>
> *Test samples in $\mathcal{D}_{\text{test}}^k$ are assumed to arrive sequentially from a time-varying and possibly shifting test distribution $(\boldsymbol{x}_k, y_k) \sim P_k, k \ge 1$. We do not make any assumptions about the nature of the distribution shift.*
>
> The reason for the generality of our framework is our use of conjugate-mixture empirical Bernstein bound [1] when constructing the confidence sequence for the test risk (see Appendix B.1 in the paper for more details).
>
> In our experiment, we report the results on time-varying shifts. For example, in Section 5.1 we report ImageNet-C results when the severity gradually increases from severity 0 to severity 5 across the test stream, whereas in Section 5.2 we report results for Yearbook and FMoW-Time datasets, both of which contain time-varying (i.e., non-stationary) shifts [4].
>
> > [...] whole ImageNet-C with the CoTTA proposed setting may better prove its effectiveness.
>
> Here we assume you are asking how our monitoring tool performs on a test stream where  different ImageNet-C corruptions are applied sequentially. If we have misunderstood your point, please kindly let us know.
>
> **During the rebuttal, we have tested our risk monitoring framework on a test stream where all ImageNet-C corruptions were sequentially appended at severity level 5. As seen in the results below, the effectiveness of our monitoring tool generalizes to this stream as well.** This is a direct consequence of the aforementioned generality of our theoretical framework where we do not make any assumptions on the nature of the distribution shift.
>
> Note that since we cannot attach figures to the rebuttal, the results are summarised using 2 quantitative metrics: (i) average detection delay and (ii) time-average of Assumption 1 delta (see line 284 in our manuscript for the exact definition).
>
> ### ImageNet-C, Gaussian Noise→…→ JPEG, Severity 5
>
> |  | Avg detection delay ($\downarrow$) | Assumption 1 delta ($\downarrow^+$) |
> | --- | --- | --- |
> | SOURCE | 33 | 0.00 |
> | TENT | 108 | 0.08 |
> | SAR | 91 | 0.05 |
> | SHOT | 96 | 0.05 |
> | COTTA | 52 | 0.12 |
>
> ### ImageNet-C, Gaussian Noise, Severity 0→5 (Figure 7)
>
> |  | Avg detection delay ($\downarrow$) | Assumption 1 delta ($\downarrow^+$) |
> | --- | --- | --- |
> | SOURCE | 297 | 0.05 |
> | TENT | 269 | 0.03 |
> | SAR | 424 | 0.04 |
> | SHOT | 237 | 0.04 |
> | COTTA | 241 | 0.04 |
>
> > Limited Collapse Prevention Comparisons:  [...]
>
> We would like to clarify that the primary goal of our risk monitoring tool is to detect when the test risk (i.e., error) exceeds a predetermined performance threshold.  Wee see this is a distinct task from collapse *prevention*, which we interpret as either (i) some form of on-going regularization that prevents collapse, or (ii) a method that allows TTA to recover once collapse has already occurred.  In either case, our contribution is complementary since our method can detect when (i) fails to prevent collapse or, in the case of (ii), trigger an prevention strategy (e.g. rolling back to the non-collapsed model).
>
> Moreover, note that risk monitoring is a more general goal than collapse *detection*, as performance degradation can occur even in the absence of collapse (e.g., under harmful distribution shifts). As such, we found it more meaningful to compare our method with existing risk monitoring tools [5], and we presented an extended comparison to relevant baselines in Appendix A.5. It is true that in Section 5.3 we discuss model collapse in the context of TENT. However, this was not intended to frame our tool as a collapse detector. Rather, our aim was to demonstrate that our monitoring tool remains effective even when the underlying TTA method collapses—a non-trivial result given that our monitoring relies on the model’s own uncertainty estimates.
>
> > Unquantified Computational Overhead:  [...]
>
> We agree that minimizing the computational overhead introduced by monitoring is important. To this end, we would like to highlight that our monitoring “wrapper” is lightweight. As shown in Algorithm 1 in Appendix C, the main additional cost of monitoring comes from threshold calibration (Algorithm 2) which involves running the adapted model on the calibration data—an operation that can typically be performed in a single forward pass thanks to the small size of the calibration set (see Appendix A.4)—followed by computing F1 scores over a fixed grid of candidate thresholds. As a result, the runtime of our monitoring component is small compared to the TTA block, which often requires a backward pass to update the model. We will include a more detailed discussion of the computational costs in the camera-ready version, along with precise timing measurements.
>
> As part of the rebuttal, we also conducted an experiment where threshold calibration is performed not at every step (as in the main experiments reported in the paper), but only every $S>1$ steps, which naturally reduces the computational overhead. As shown in the table below, monitoring remains effective even for $S > 1$. We will add these results to Appendix A.4.
>
> ### ImageNet-C Sev 0→5
>
> |  | Avg. detection delay ($\downarrow$) | Assumption 1 delta ($\downarrow^+$)  |
> | --- | --- | --- |
> | TENT (S=1) | 269 | 0.03 |
> | TENT (S=10) | 269 | 0.03 |
> | TENT (S=20) | 280 | 0.04 |
>
> > How does the framework perform when Assumption 1 is persistently violated? [...]
>
> Throughout our experiments, we have not encountered a scenario where Assumption 1 is persistently violated. According to our theoretical analysis (Section 3.2), if such a persistent violation were to occur, the lower bound would no longer be valid, and our monitoring tool could begin raising false alarms at a rate exceeding the guaranteed level $\alpha$. As noted in Section 5.2, we have observed small violations of Assumption 1 for small values of $t$ on the FMoW-Time dataset. Fortunately, at small $t$, the finite-sample penalty term in the confidence sequence is largest, which may help compensate for these violations and reduce the risk of false alarms.
>
> On the flip side, a more practical issue we have observed is that Assumption 1 can sometimes be satisfied “too much,” resulting in a lower bound that is overly loose. This occurs, for instance, when using model uncertainty as a loss proxy with last-layer TTA methods, due to the normalization applied to the classification head (see Section 5.4). To address this, we propose using the distance to the closest class prototype as a more suitable loss proxy for such methods, which leads to tighter and more informative bounds.
>
> > Can the framework be tuned to prioritize early detection over false alarm rate control […]
>
> > Detection Delay: […]
>
> We agree that the detection delays introduced by our monitoring tool may be too large for scenarios where late detection is significantly more costly than false alarms, as we also note in the limitations section in Section 6. As discussed in Appendix A.1, such delays are to some extent unavoidable when aiming to guarantee a low probability of false alarm (PFA), as is the case in our work. That’s because a PFA guarantee necessitates working with the *lower* confidence sequence on the true test risk (error), which itself must be further lower-bounded in our unsupervised setting (see Proposition 1). This “double” lower-bounding results in a conservative behavior of the monitoring tool and can lead to increased detection delays.
>
> To answer the question on prioritizing the early detection instead, we believe the most promising approach is to relax the PFA guarantee and instead adopt guarantees based on the average run length (ARL). In short, unlike PFA, which controls false alarm *with high probability*, ARL controls the false alarm *in expectation*. A successful example of relying on ARL guarantees (instead of PFA) to improve detection times can be found in the literature on change-point detection [2].  As we mention in Section 6, extending our proposed TTA monitoring framework to incorporate ARL-style guarantees is a highly promising direction for future work.
>
> > Label Requirement: [...] assumption (Assumption 1) needs labeled data.
>
> Due to character-limit for the rebuttal response, please see our response to RuQX.
>
> > requirements for D_cal from the source domain may violate the source-free principle to some extent.
>
> We agree that our use of labeled calibration data from the source distribution for threshold calibration (Section 3.4) could be viewed as a minor deviation from the source-free principle. However, as demonstrated in Appendix A.4, effective risk monitoring can still be achieved with as few as 100 calibration samples. We believe this is a reasonable trade-off for the added safety afforded by rigorous performance monitoring enabled by our method. Crucially, our monitoring approach does not require any labels from the test data, maintaining full compatibility with the TTA paradigm.
>
> We thank you again for your efforts in reviewing our work. If you feel like we have sufficiently addressed your concerns, we would appreciate you raising your score.
>
> [1] Howard et al. Time-uniform, nonparametric, nonasymptotic confidence sequences. The Annals of Statistics, 2021
>
> [2] Shekhar et al. Sequential changepoint detection via backward confidence sequences. ICML 2023
>
> [3] Rosenfeld et al. (Almost) Provable Error Bounds Under Distribution Shift via Disagreement Discrepancy. NeurIPS 2023
>
> [4] Yao, Huaxiu, et al. Wild-time: A benchmark of in-the-wild distribution shift over time. NeurIPS 2022
>
> [5] Amoukou et al. Sequential Harmful Shift Detection Without Labels. NeurIPS 2024

---

> > ### Comment · Reviewer_PuqZ · 2025-08-05
> >
> > I thank the authors for their detailed response. It has successfully addressed several of my concerns, particularly regarding the framework's performance in dynamic environments and the overhead of the monitoring tool. The new experimental results on the continual ImageNet-C setting are a valuable addition that I believe should be included in the final version. While I still harbor some concerns regarding Assumption 1 and the reliance on labeled data, as these factors may restrict the real-world applicability. Given the improved clarity of your research, I intend to maintain my borderline accept rating while increasing my confidence in this assessment.

---

> ### Author Response · Authors · 2025-08-06
>
> Dear Reviewer PuqZ,
>
> Thank you for your continued engagement. We are glad we could resolve your concerns regarding (i) the applicability of our framework to non-stationary distributions and (ii) computational overhead. We will include results from the continual ImageNet-C setting in the updated version.
>
> > *Assumption 1*
>
> As noted in our response to Reviewer RuQX (also copied below for completeness, as we could not elaborate on this point in our original rebuttal due to the character limit), theoretical guarantees under distribution shift are fundamentally impossible without assumptions [1]. The goal, then, is to rely on assumptions that are valid in practical scenarios. We would also like to highlight that one of our paper’s contributions is in fact the relaxation of the assumption made in prior work [2] (see Section A.1). As a consequence, our experiments (Sections 5.1 and A.5) show that our relaxed assumption holds in most practical scenarios, whereas the stricter assumption in [2] is frequently violated.
>
> **Response to Reviewer RuQX:**
>
> *We agree that the unverifiability of Assumption 1 at deployment time—due to the lack of labels—is a limitation of our monitoring framework. We explicitly acknowledge this in Section 6, where we also highlight the development of unsupervised diagnostics for detecting potential violations of Assumption 1 as an important direction for future work.*
>
> *However, to the best of our knowledge, there is currently no method that can estimate test performance under unknown distribution shifts without labels, and provide theoretical guarantees without relying on some form of assumption. As noted in [1]: “Bounding error under distribution shift is fundamentally impossible without assumptions.” As such, we believe the real question is not how to have an assumption-free monitoring tool, but rather what is the assumption that (i) is satisfied across most TTA scenarios and (ii) yields an effective monitoring signal. We believe our empirical results demonstrate that Assumption 1 successfully satisfies both of these desiderata.*
>
> > *Labeled source calibration set*
>
> While the use of a labeled source calibration dataset might slightly deviate from the TTA principle, it is not uncommon for TTA methods to rely on labeled source samples. For completeness, we list below a few examples of TTA methods that rely on labeled source data and refer to A.4 for further examples. In A.4, we show that $N\_{cal}=100$ is sufficient for our framework to detect risk violations in a timely manner on ImageNet.
>
> - EcoTTA [3]: uses 10%–20% of labeled source data (128K–256K for ImageNet) to train a meta-network
> - CAFA [4]: uses 5% of labeled source data (64K for ImageNet) to fit the prototype distribution
> - SWR+NSP [5]: uses 1K labeled source samples to align prototypes
>
> Thank you again for your valuable feedback. We remain available for any follow-up questions.
>
> [1] Rosenfeld et al. (Almost) Provable Error Bounds Under Distribution Shift via Disagreement Discrepancy. NeurIPS 2023
>
> [2] Amoukou et al. Sequential Harmful Shift Detection Without Labels. NeurIPS 2024
>
> [3] Song et al. Ecotta: Memory-efficient continual test-time adaptation via self-distilled regularization. CVPR 2023
>
> [4] Jung et al. Cafa: Class-aware feature alignment for test-time adaptation. ICCV 2023
>
> [5] Choi et al. Improving test-time adaptation via shift-agnostic weight regularization and nearest source prototypes. ECCV 2022

---

### Official Review · Reviewer_cUHi · 2025-07-03

**Clarity:** 3
**Significance:** 2
**Originality:** 2
**Rating:** 5
**Confidence:** 4

**Summary:**

The paper proposes to detect the degradation in test time adaptation models over time. The core idea is to adapt the sequential testing technique for risk monitoring to the case of time-varying models and unlabeled data.

**Questions:**

None

**Ethical Concerns:**

["NO or VERY MINOR ethics concerns only"]

**Final Justification:**

The rebuttal addresses my concern and I raise my rating to accept.

**Limitations:**

Yes

**Quality:**

3

**Strengths And Weaknesses:**

**Strengths:**
- Previous risk monitoring techniques such as sequential testing was designed for static models on labeled data. The authors extend it to handle the test-time adaptation case. The key is to monitor the losses of a sequence of models updated during the adaptation, and to design proxy loss functions to eliminate the need for labels.
- The proposed method is straight-forward and easy to implement.

**Weaknesses:**
- In the paper only the classification task is considered. It is unclear for other tasks whether we can find suitable loss proxies and whether this method is still effective.
- This framework does not handle label distribution shift, where only the label distribution is changing over time, but not the input distribution.

---

> ### Author Rebuttal · Authors · 2025-07-30
>
> We thank you for your time and efforts. We address your points below.
>
> > *In the paper only the classification task is considered. It is unclear for other tasks whether we can find suitable loss proxies and whether this method is still effective.*
>
> To the best of our knowledge, classification is (by far) the most studied task in the test-time adaptation literature [1], which is why we focused on it in our experiments. That said, our theoretical framework is general and should be readily extendable to other tasks such as regression by finding the suitable loss proxies. We view this as a highly promising direction for future work. If there are some specific tasks you have in mind, please let us know, and we would be happy to discuss how our framework could be adapted accordingly.
>
> > *This framework does not handle label distribution shift, where only the label distribution is changing over time, but not the input distribution.*
>
> **We would like to clarify that our theoretical framework does handle the case of a shifting label distribution** $P_k(y)$. As we write in Section 2 (lines 55-57),
>
> *Test samples in* $\mathcal{D}_{\textrm{test}}^k$ *are assumed to arrive sequentially from a time-varying and possibly shifting test distribution* $(x_k, y_k) \sim P_k, k \ge 1$. *We do not make any assumptions about the nature of the distribution shift.*
>
> During the rebuttal we experimentally validated our framework when the label distribution $P_k(y)$ is changing. For that, we order Yearbook test samples according to their class labels to induce label shift per batches, as in [2]. Due to our inability to attach plots, we summarize the results in the table format below. We report (i) average detection delay which is defined as the difference between the time at which our unsupervised lower bound crosses the performance threshold $\hat{t}\_{\text{min}}^b$ and the time at which the “true” test risk does so $\hat{t}_{\text{min}}^a$, and (ii) the time-average of Assumption 1 delta (as defined in line 284 of the manuscript) which directly reflects the tightness and validity of our bound. **Results below show that our risk monitoring framework is also effective at detecting risk violation that arise due to the presence of label shift.** We will include those results to the camera-ready version and would like to thank you for this suggestion.
>
> ### Yearbook (without label shift, Figure 3):
>
> |  | Avg detection delay $(\downarrow)$ | Assumption 1 delta $(\downarrow^+)$ |
> | --- | --- | --- |
> | SOURCE | 76 | 0.07 |
> | TENT | 84 | 0.07 |
> | SAR | 87 | 0.07 |
> | SHOT | 73 | 0.06 |
> | COTTA | 58 | 0.08 |
>
> ### Yearbook (with label shift):
>
> |  | Avg detection delay $(\downarrow)$ | Assumption 1 delta $(\downarrow^+)$ |
> | --- | --- | --- |
> | SOURCE | 66 | 0.07 |
> | TENT | 68 | 0.17 |
> | SAR | 71 | 0.18 |
> | SHOT | 39 | 0.15 |
> | COTTA | 35 | 0.17 |
>
> If we have clarified your concerns, we would appreciate your consideration in raising your score.
>
> [1] Xiao, Zehao, and Cees GM Snoek. Beyond model adaptation at test time: A survey. Arxiv 2024.
>
> [2] Gong, Taesik, et al. NOTE: Robust continual test-time adaptation against temporal correlation. NeurIPS 2022

---

> ### Author Response · Authors · 2025-08-05
>
> Dear Reviewer cUHi,
>
> we thank you again for your reviewing effort. Since the discussion period is slowly approaching the end, we were wondering if you had a chance to look at our rebuttal?
>
> We remain available for any potential follow-up questions.
>
> Kind regards,
>
> Authors

---

### Official Review · Reviewer_RuQX · 2025-07-07

**Clarity:** 3
**Significance:** 2
**Originality:** 2
**Rating:** 4
**Confidence:** 4

**Summary:**

This paper proposes a method for monitoring performance during test-time adaptation (TTA) without access to test labels. By extending sequential testing with confidence sequences, it develops unsupervised alarms based on loss proxies (specifically, model uncertainty), enabling reliable detection of performance degradation. The method is supported by theoretical analysis and is empirically validated across datasets, shift types, and TTA strategies.

**Questions:**

Questions
- Is there a practical way to diagnose whether Assumption 1 holds at deployment time?
- How sensitive is the method to hyperparameters (e.g., ϵ_tol, α, λ, τ)?

**Ethical Concerns:**

["NO or VERY MINOR ethics concerns only"]

**Final Justification:**

This paper presents a sound, yet incremental, contribution to TTA. I am not opposed to accepting it, but I would not actively advocate for its acceptance either.

**Limitations:**

Yes

**Quality:**

2

**Strengths And Weaknesses:**

Strenghts

- Important Problem: The paper tackles a critical challenge in test-time adaptation, due to risks such as silent model collapse or performance degradation, which current TTA methods often fail to detect.

- Practical Proxy: It leverages predictive uncertainty as a simple and efficient proxy for model error, requiring no auxiliary components. The use of online threshold calibration ensures effectiveness even under continuous adaptation.

- Broad Applicability: The proposed method seems to generalize well across TTA methods, datasets, and types of distribution shift.

Weakness:

- Unverifiable Assumption: The method relies on Assumption 1 (proxy informativeness) for its guarantees, but this assumption cannot be verified without labels, limiting trust in fully unsupervised deployments.

- Proxy Sensitivity: Performance depends on the choice of proxy and threshold calibration. For example, standard uncertainty fails for last-layer TTA methods under severe distribution shifts.

- Hyperparameter Robustness: The method introduces several hyperparameters (ϵ_tol, α, λ, τ), but their sensitivity to different shifts is not analyzed. Given that TTA methods are known to be HP-sensitive [1,2], it's important to study how these choices affect risk detection.

[1] Parameter-Free Online Test-Time Adaptation, CVPR 2022 \
[2] On Pitfalls of Test-Time Adaptation, ICML 2023

---

> ### Author Rebuttal · Authors · 2025-07-30
>
> We thank you for your feedback. We address your questions below.
>
> > *Hyperparameter Robustness: The method introduces several hyperparameters (ϵ_tol, α, λ, τ), but their sensitivity to different shifts is not analyzed. Given that TTA methods are known to be HP-sensitive [1,2], it's important to study how these choices affect risk detection.*
>
> > *How sensitive is the method to hyperparameters (e.g., ϵ_tol, α, λ, τ)?*
>
> We would like to clarify that the parameters mentioned ($\epsilon\_{\text{tol}}$, $\alpha$, $\lambda$, $\tau$) are not hyperparameters. Instead, they are either user-specified parameters that control the conservativeness of the statistical testing framework ($\alpha$, $\epsilon\_{\text{tol}}$), or thresholds that are *automatically tuned* by our proposed online calibration procedure ($\lambda$, $\tau$). Below, we summarize how each parameter is defined in the manuscript:
>
> - User-specificed parameters:
>
>     - $\alpha$: This is the significance level used in the sequential test (Section 3.1). As with standard statistical tests, this parameter is chosen by the user based on task-specific requirements. A lower $\alpha$ makes the test more conservative—prioritizing Type I error control (i.e., reducing false alarms) at the cost of reduced power (i.e., lower detection sensitivity).
>     - $\epsilon\_{\text{tol}}$: This is a parameter that determines what constitutes the performance drop at test time that is still acceptable. For $\epsilon\_{\text{tol}} = 0$, the user wants to trigger the alarm as soon as the performance on test data gets worse compared to the performance on the source data, whereas for $\epsilon\_{\text{tol}} > 0$ the user allows for some ‘slack’ at test time.
>
> We consider the ability to accommodate a range of values for $\alpha$ and $\epsilon\_{\text{tol}}$ a strength of our framework, as this means that our monitoring framework is applicable across various settings with different requirements on when to trigger the alarm.
>
> - Automatically estimated thresholds:
>
>     - $\lambda\_k$ and $\tau$: As described in Section 3.2, $\lambda\_k$ is the loss proxy threshold at time step $k$, and $\tau$ is the loss threshold. Both are estimated online using our calibration procedure (see Section 3.4). Hence, they are not fixed hyperparameters but ‘fitted’ parameters of our monitoring tool.
>
> Importantly, unlike prior threshold calibration approaches [1], which require manual hyperparameter tuning (e.g., an FDP cut-off in [1], see Section 4.1 there), our threshold finding procedure does not introduce any hyperparameters. **As such we find that our *hyperparameter-free* threshold calibration generalizes across a large set of TTA methods and shift types in our experiments (see Section 5.2).**
>
> > *Proxy Sensitivity: Performance depends on the choice of proxy and threshold calibration. For example, standard uncertainty fails for last-layer TTA methods under severe distribution shifts.*
>
> We agree that selecting an appropriate loss proxy is crucial for the effectiveness of our proposed risk monitoring framework. However, we would like to emphasize that our choice of model uncertainty as a proxy has demonstrated robustness, generalizing well across five TTA methods and four types of distribution shifts (see Section 5.2). While it does indeed underperform for last-layer TTA methods—as we acknowledge ourselves in Section 5.4—this behavior is expected. As we explain there, last-layer TTA methods normalize the classification head, making uncertainty less informative of the true loss. To address this, we propose an alternative proxy based on the distance to class prototypes, which we found to be effective (see Figure 5). Our motivation for pointing out the failure of model uncertainty in the case of last-layer methods was to illustrate the flexibility of our risk monitoring framework to accommodate different loss proxies, and to highlight that the choice of the proxy should be informed by the characteristics of the given TTA method.
>
> Regarding the dependence on threshold calibration, we found that our proposed online calibration procedure (Section 3.4) generalizes well across all TTA/shift scenarios considered in our experiments—including last-layer methods. As such, we are unsure about the concern raised here. If you could elaborate further, we would be happy to provide additional clarification or conduct further experiments.
>
> > *Unverifiable Assumption: The method relies on Assumption 1 (proxy informativeness) for its guarantees, but this assumption cannot be verified without labels, limiting trust in fully unsupervised deployments.*
>
> > *Is there a practical way to diagnose whether Assumption 1 holds at deployment time?*
>
> We agree that the unverifiability of Assumption 1 at deployment time—due to the lack of labels—is a limitation of our monitoring framework. We explicitly acknowledge this in Section 6, where we also highlight the development of unsupervised diagnostics for detecting potential violations of Assumption 1 as a important direction for future work.
>
> However, to the best of our knowledge, there is currently no method that can estimate test performance under unknown distribution shifts without labels *and* provide theoretical guarantees without relying on some form of assumption. As noted in [2]: *"Bounding error under distribution shift is fundamentally impossible without assumptions."* As such, we believe the real question is not how to have an assumption-free monitoring tool, but rather what is the assumption that (i) is satisfied across most TTA scenarios and (ii) yields an effective monitoring signal. We believe our empirical results demonstrate that Assumption 1 successfully satisfies both of these desiderata.
>
> We thank you again for reviewing our work. If you feel like we have sufficiently addressed your concerns, we would appreciate you raising your score.
>
> [1] Amoukou et al. Sequential Harmful Shift Detection Without Labels. NeurIPS 2024
>
> [2] Rosenfeld et al. (Almost) Provable Error Bounds Under Distribution Shift via Disagreement Discrepancy. NeurIPS 2023

---

> ### Author Response · Authors · 2025-08-05
>
> Dear Reviewer RuQX,
>
> we thank you again for your reviewing effort. Since the discussion period is slowly approaching the end, we were wondering if you had a chance to look at our rebuttal?
>
> We remain available for any potential follow-up questions.
>
> Kind regards,
>
> Authors

---

### Note · Authors · 2025-08-12

Dear AC,

Thank you for your time and effort.

We are encouraged that, based on our rebuttal, **the reviewers reached a consensus on recommending acceptance of our work**. From our understanding, the only remaining concerns cited by the reviewers in their replies as reasons for not recommending acceptance more strongly are: (i) Assumption 1 cannot be verified without labels (PuqZ), and (ii) the perceived limited novelty of our work (LLXY).

For (i), as already mentioned in our rebuttal, we consider it a strength that the assumption required for our monitoring to be valid is explicitly stated—unlike many TTA approaches, where the exact assumptions underlying a method's validity are often left unspecified. Moreover, we have shown that in practice Assumption 1 holds across a diverse range of datasets, shift types, and TTA methods, demonstrating that it is far from restrictive in typical TTA settings. Finally, we would like to emphasize that the task we address, i.e., **monitoring test performance without labels under arbitrary distribution shifts and model adaptations**, is inherently challenging, and it is almost inevitable that some assumptions are necessary if one aims to achieve any form of theoretical guarantees (such as our PFA control). No prior work on unsupervised risk monitoring is entirely assumption-free, and all other existing approaches also rely on assumptions that require labeled data for verification; see, for example, [1, 2].

For (ii), we maintain that our work represents a novel, technically non-trivial, and important contribution to the field of TTA. This has been acknowledged by almost all reviewers (PuQZ: "*Novel Unsupervised Monitoring Approach: … The paper fills a significant gap by risk monitoring for TTA without test labels*"; d5NY: "*This paper is the first one (that I know of) that deals with the risk monitoring in the test-time adaptation*"), including LLXY themselves: "*This paper presents a novel unsupervised risk monitoring framework for test-time adaptation (TTA), addressing the critical challenge of detecting performance degradation*." Additionally, what made it hard for us to respond to LLXY's novelty concerns is that they failed to provide any concrete references to back-up their claims, despite our repeated requests.

[1] Amoukou et al. Sequential Harmful Shift Detection Without Labels. NeurIPS 2024

[2] Rosenfeld et al. (Almost) Provable Error Bounds Under Distribution Shift via Disagreement Discrepancy. NeurIPS 2023

---

### Decision · Program_Chairs · 2025-09-17

**Decision:**

Accept (poster)

**Comment:**

This paper proposes a sequential testing method for monitoring the risk of performance degradation in Test-Time Adaptation (TTA) models, assuming one has good proxies of the loss, but not necessarily assuming one has access to the ground truth labels themselves. Reviewers agreed that this paper addresses an important problem, appreciated the theory around the paper (though highlighted some concerns about its practicality), and applauded the comprehensive testing across a wide variety of datasets. The exploration and testing of different proxy measures are useful contributions to the AI monitoring field.

I recommend acceptance of the work but with three major requests for the authors. First, as pointed out by reviewers, Assumption 1 is very strong (must uniformly hold over time) and cannot be verified. The authors demonstrate that Assumption 1 approximately holds in semi-synthetic image datasets and two real-world image datasets, but it's difficult to know if this assumption would similarly hold in other datasets (e.g. tabular). I would suggest testing out the method and checking if Assumption 1 holds in other settings as well. Also, it would be important to provide some general guidance on when Assumption 1 can be expected to hold. Second, I suggest the authors discuss the relation of this work to other comparator methods (and include comparisons when appropriate). For instance, works such as monitoring AI algorithms under performativity also account for AI algorithms that learn over time (see e.g. Feng et al 2024 i AISTATS). Finally, comparators in the simulated section should also be included in the real-world datasets, because the real-world datasets used in these experiments also have true labels. This will help gauge how much slower the monitoring methods are due to the use of proxies.